# Towards Unified Benchmark and Models for Multi-Modal Perceptual Metrics

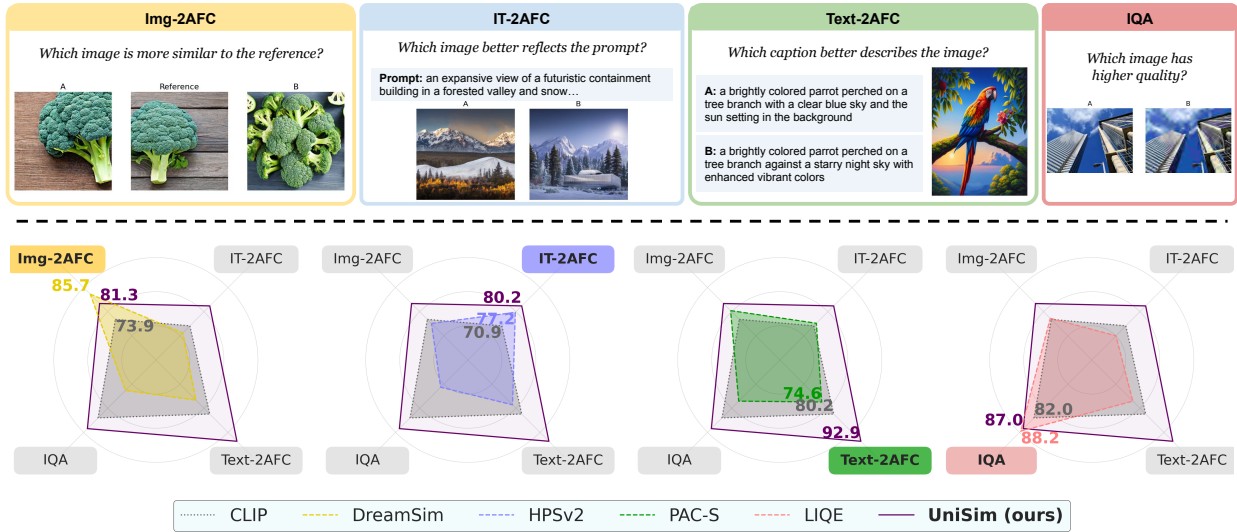

Figure 1: **Summary of our UniSim framework. (1)** We unify existing multi-modal perceptual similarity tasks into a single comprehensive benchmark **UniSim-Bench** (whose *Core 2AFC Tasks* are illustrated in the top row). **(2)** We show that models specialized in individual tasks (e.g., DreamSim (Fu et al., 2023), HPSv2 (Wu et al., 2023b), PAC-S (Sarto et al., 2023), LIQE (Zhang et al., 2023b)) do not generalize well to both unseen perceptual tasks and unseen datasets within the same task, even with worse accuracy than CLIP (Radford et al., 2021). **(3)** We introduce our multi-task perceptual metric **UniSim** which surpasses the baseline CLIP model and has superior or competitive performance across tasks compared to the specialized models.

## Abstract

Human perception of similarity across uni- and multi-modal inputs is highly complex, making it challenging to develop automated metrics that accurately mimic it. While general-purpose vision-language models (VLMs) like CLIP and large multi-modal models (LMMs) can serve as zero-shot perceptual metrics, they are not explicitly trained for this task. As a result, recent efforts have developed specialized models for narrow perceptual tasks. However, the extent to which these metrics align with human perception remains unclear. To address this, we introduce UniSim-Bench, a benchmark covering seven multi-modal perceptual similarity tasks across 25 datasets. Our evaluation reveals that models fine-tuned on a specific dataset struggle to generalize to unseen datasets within the same task or to related perceptual tasks. As a first step towards a unified multi-task perceptual similarity metric, we fine-tune both encoder-based and generative vision-language models on a subset of UniSim-Bench tasks. This approach achieves the highest average performance and, in some cases, surpasses task-specific models. Our comparative analysis demonstrates that encoder-based VLMs exhibit superior generalization capabilities as perceptual metrics. However, these models still struggle with unseen tasks, underscoring the challenge of developing a robust, unified metric that accurately captures the human notions of similarity.

# 1 Introduction

Developing automated metrics that replicate human perception of similarity remains a complex and open problem due to its intricate and multi-dimensional nature. With the rapid advancement of vision-language models (Radford et al., 2021; Podell et al., 2023; Liu et al., 2024; Li et al., 2024b; Achiam et al., 2023), there is a growing need for metrics capable of evaluating similarity across multiple modalities. Such perceptual metrics are particularly beneficial for tasks like measuring the alignment of large autoregressive multi-modal models (LMMs) (Achiam et al., 2023; Team, 2024) and assessing the quality of text-to-image generative models (Zhang et al., 2023a). Moreover, human-aligned visual representations (achieved by fine-tuning the foundation models on perceptual data) have recently been shown to outperform non-aligned ones in certain vision (non-perceptual) downstream tasks (Sundaram et al., 2024).

Prior works (Hessel et al., 2021; Fu et al., 2023) have shown that foundation encoder models like CLIP (Radford et al., 2021) or DINO (Caron et al., 2021) can be used as expressive metrics, where the semantic similarity between visual or text inputs is approximated through the alignment of embedding vectors. Moreover, LMMs (Li et al., 2024b; Jiang et al., 2024; Achiam et al., 2023) can be prompted to solve perceptual tasks using natural language. While these models exhibit strong zero-shot performance on some perceptual tasks, they often struggle with more fine-grained or complex tasks. As a result, specializing encoder-based (Liu et al., 2021; Lee et al., 2021; Xu et al., 2023; Wu et al., 2023b; Sarto et al., 2023; Wada et al., 2024; Zhang et al., 2023b) and generative models (Wu et al., 2023a; 2024; Zhu et al., 2024b) for narrow applications, e.g., image-to-image similarity or text-image alignment, has become a relevant research direction.

Despite this progress, the extent to which current metrics generalize to unseen datasets within the same task or to related perceptual tasks—and consequently, their ability to truly capture the human notion of similarity—remains unclear. We argue that a systematic and comprehensive evaluation of existing automated metrics requires unifying existing multi-modal perceptual tasks, which prior work has largely treated as disjoint problems. In fact, they represent distinct but interconnected facets of human perception, and therefore, a unified framework is essential to holistically evaluate and develop more comprehensive perceptual metrics. As a first step, we introduce UniSim-Bench, a benchmark integrating 7 widely used uni- and multi-modal perceptual tasks (illustrated in Fig. 1 and Fig. 2), encompassing 25 datasets, in a single framework. In particular, we split the seven tasks into two categories: the first includes two-alternative forced choice (2AFC) tasks (Link and Heath, 1975), offering a set of complementary uni- and multi-modal similarity tasks, while the second group aim to assess perceptual understanding in more diverse scenarios. Our evaluation on UniSim-Bench reveals significant limitations of current perceptual metrics. We observe **limited intra-task generalization**, where models fine-tuned on a specific dataset often struggle to generalize to other datasets within the same task. Additionally, there is **poor inter-task generalization**, i.e., the good performance of specialized metrics does not transfer to strongly correlated tasks (Fig. 3). For instance, metrics specialized in IT-2AFC do not effectively transfer to Text-2AFC, despite both tasks assessing image-text alignment. These weaknesses highlight the gaps of the current model in capturing human perception and may limit their applicability.

To address these limitations, we propose UniSim, a family of unified multi-task perceptual models. We fine-tune both CLIP (Radford et al., 2021) and LLaVA-NeXT (Li et al., 2024b) on multiple perceptual datasets using tailored multi-task learning approaches. The UniSim models achieve higher average accuracy across tasks than the baselines and exhibit generalization to left-out datasets within each task, showing the viability of a unified perceptual metric. By systematically ablating training data (Table 4), we find evidence of cross-task generalization. Moreover, by extensive comparison between CLIP-based and LMM-based UniSim models we found out that the superior generalization capabilities of encoder-based models. For out-of-distribution (OOD) perceptual tasks, UniSim demonstrates improvements on unseen tasks with structures similar to the training ones. However, generalizing to more diverse uni- and multi-modal tasks remains a significant challenge. In summary, the main contributions of our work, also illustrated in Fig. 1, are:

- **UniSim-Bench**, the first benchmark to track the progress of perceptual similarity metrics across uni- and multi-modal tasks, including both core 2AFC tasks and a diverse set of tasks to test out-of-distribution generalization,

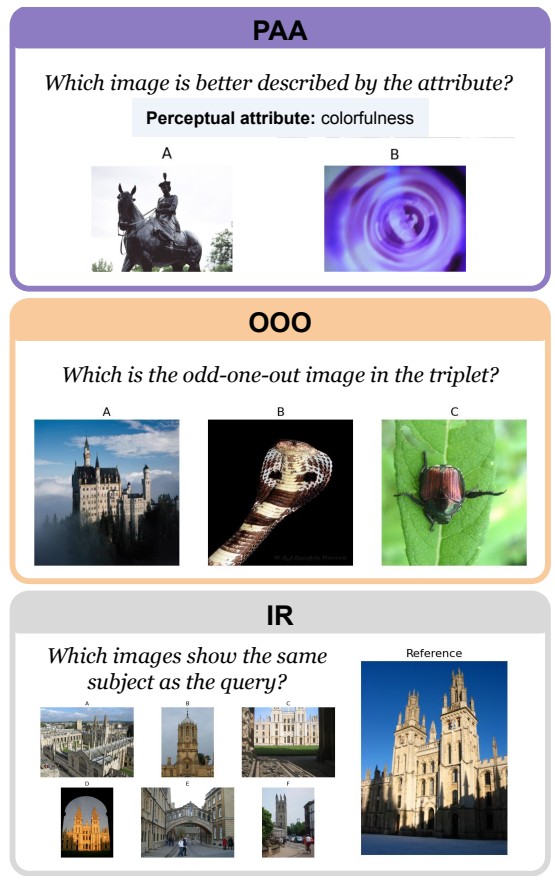

Figure 2: *OOD Generalization Tasks* in **UniSim-Bench.** We illustrate samples from the three tasks not used for training, to evaluate the model's generalization capabilities.

Table 1: **Composition of UniSim-Bench.** We detail the datasets used to evaluate each task in our benchmark, as well as whether they are used to train our UniSim models.

| Task | Dataset | UniSim trains on | Test samples |
|---|---|:---:|---:|
| *Core 2AFC Tasks* | | | |
| Img-2AFC | Nights | ✓ | 1824 |
| | Bapps | ✗ | 5K |
| | PieAPP | ✓ | 3314 |
| IT-2AFC | ImageReward | ✗ | 412 |
| | HPDv2 | ✓ | 5K |
| | Agiqa-3k | ✗ | 5K |
| | MagicBrush | ✓ | 693 |
| | HQ-Edit | ✓ | 2K |
| Text-2AFC | $\mathcal{CD}$-Coco | ✗ | 780 |
| | Polaris | ✓ | 5K |
| | HQ-Edit | ✓ | 2K |
| IQA | Kadid-10k | ✓ | 5K |
| | KonIQ-10k | ✗ | 5K |
| | PieAPP | ✓ | 5K |
| | Agiqa-3k | ✗ | 5K |
| | Pipal | ✓ | 3025 |
| *OOD Generalization Tasks* | | | |
| PAA | Sice | ✗ | 2151 |
| | KonIQ-10k | ✗ | 4 x 5K |
| OOO | Cifar-100-OOO | ✗ | 5K |
| | ImageNet-OOO (ours) | ✗ | 5K |
| IR | $\mathcal{R}$Oxford | ✗ | 70 |
| | $\mathcal{R}$Paris | ✗ | 70 |
| **Total** | | | **88K** |

- **Identification of the limitations of current specialized perceptual metrics** in generalizing to unseen datasets and perceptual tasks, providing deeper insights into the shortcomings of existing metrics,

- **UniSim**, a set of multi-task perceptual models trained on a curated dataset covering the *Core 2AFC Tasks*, serving as a first step toward general-purpose perceptual metrics,

- **Analyses** of the interaction between different tasks during training and of the generalization capabilities of encoder- versus generative-based VLMs.

Together, UniSim-Bench and UniSim open the way towards understanding the challenges of learning automated metrics that broadly mimic human perceptual similarity, beyond narrow, task-specific applications.

## 2 Background and Related Work

In the following, we provide background on perceptual similarity tasks and metrics, along with the most relevant related works (an extended discussion is available in App. A).

## 2.1 Perceptual similarity tasks

Learning to assess the similarity between data items in a way that aligns with human perception has long been a core challenge in computer vision and machine learning. Traditional perceptual metrics often focused on uni-modal tasks, e.g., assessing image-to-image similarity (Zhang et al., 2018; Fu et al., 2023) or quality in denoising and compression contexts (Wang et al., 2023; Zhang et al., 2023b). Recent advances in generative and multi-modal AI call however for perceptual metrics addressing cross-modal consistency, as they are used for training and evaluating text-to-image generative models (Xu et al., 2023; Wu et al., 2023b), captioning models (Li et al., 2023b; 2024b), and the perceptual capabilities of multi-modal LLMs (Li et al., 2024b; Jiang et al., 2024). Despite shared goals, prior work has generally treated these perceptual tasks as isolated problems, and developed distinct approaches. To fill this gap, we propose a unified framework that enables consistent evaluations of existing metrics and the development of generalized perceptual similarity metrics across uni- and multi-modal domains.

## 2.2 Foundation models as perceptual metrics

**Encoder models.** Replacing raw data with deep features extracted from pre-trained neural networks has become the standard in perceptual metrics. These learned representations can better capture human-perceived similarity compared to traditional metrics, and are used in tasks like image-to-image similarity (Zhang et al., 2018; Liu et al., 2021; Croce et al., 2025), text-image alignment (Hessel et al., 2021; Lee et al., 2021; Xu et al., 2023; Sarto et al., 2023; Wada et al., 2024), image quality assessment (Wu et al., 2023b; Zhang et al., 2023b). Foundation models like CLIP (Radford et al., 2021) and BLIP (Li et al., 2023b) have been the basis for many of these metrics. Specifically, CLIP consists of an image encoder, $\phi : I \to \mathbb{R}^D$, and a text encoder $\psi : T \to \mathbb{R}^D$, which project data from different modalities into a shared $D$-dimensional latent space. Using contrastive learning, CLIP aligns the embeddings of image-text pairs with their corresponding semantic meanings within this latent space. The similarity between inputs can be then quantified by the cosine similarity of their embedding vectors. For instance, given a caption $\boldsymbol{t} \in T$ and two images $\boldsymbol{x}_1, \boldsymbol{x}_2 \in I$, a CLIP model can determine which image better aligns with the caption by solving: $\arg\max_{\boldsymbol{z} \in \{\boldsymbol{x}_1, \boldsymbol{x}_2\}} \mathtt{sim}_{\phi,\psi}(\boldsymbol{z}, \boldsymbol{t})$, where $\mathtt{sim}_{\phi,\psi}(\boldsymbol{z}, \boldsymbol{t}) = \left\langle \frac{\phi(\boldsymbol{z})}{\|\phi(\boldsymbol{z})\|_2}, \frac{\psi(\boldsymbol{t})}{\|\psi(\boldsymbol{t})\|_2} \right\rangle$ is the generic similarity function that uses the CLIP encoders $\phi, \psi$ to measure the similarity of the items of any input pair (in this case an image-text pair). Encoder models have the advantage of associating each input with a single feature vector, allowing reuse for multiple comparisons. This leads to efficient evaluation across several images and texts without redundant computations. This efficiency is especially valuable in tasks like retrieval, where the similarity of a query must be measured against hundreds or thousands of references.

**Generative models.** Recently, large multi-modal models (LMMs) have made significant progress (Liu et al., 2024; Ye et al., 2024; Ghazanfari et al., 2024a; Wang et al., 2024b; Chen et al., 2024), achieving strong capabilities in multi-image understanding and reasoning (Li et al., 2024b; Jiang et al., 2024; Ye et al., 2024). This makes LMMs promising alternatives to traditional encoders as perceptual metrics. A generalist LMM can be easily adapted to specific perceptual tasks using prompting. In the example above, one could query `"Image A: <`$\boldsymbol{x}_1$`>, Image B: <`$\boldsymbol{x}_2$`>. Which image is better described by <`$\boldsymbol{t}$`>?"`. This approach offers greater flexibility than encoder models, leveraging the extensive training and scale of these LMMs. However, a key drawback is the challenge of scaling LMMs to tasks involving many text prompts or images, such as image-to-image retrieval. In addition to generalist models (Li et al., 2024b; Jiang et al., 2024), some works have specialized LMMs for specific perceptual tasks, often focusing on single-image evaluations, such as image quality assessment (Wu et al., 2023a; 2024; Zhu et al., 2024b; You et al., 2025), and image aesthetics evaluation (Huang et al., 2024).

## 2.3 Benchmarks

Several benchmarks have been recently developed to evaluate the perceptual and multi-modal understanding capabilities of vision-language models. BLINK (Fu et al., 2024) covers 14 visual perception tasks, but includes only a single dataset for image-to-image similarity. MUIRBENCH (Wang et al., 2024a) assesses 12 multi-image understanding tasks, with one about image-text alignment. Also about image-text similarity,

several benchmarks (Ku et al., 2023; Li et al., 2024a; 2023a; Zhang et al., 2024b) offer comprehensive frameworks for evaluating text-to-image generative models. In visual quality analysis, Q-Bench, Q-Bench+ (Wu et al., 2023a), 2AFC-LMM (Zhu et al., 2024a), and MICBench (Wu et al., 2024) assess a wide range of visual attributes, including low-level perception, detailed description, and overall quality. While each of these benchmarks addresses some particular facets of perceptual evaluation, they often focus on reasoning and understanding tasks. This underscores further the need for a comprehensive benchmark to assess the perception capabilities of automated metrics across all aspects of multi-modal similarity.

## 3 Towards a Unified Framework for Multi-Modal Perceptual Similarity Tasks

We here introduce our unified framework for benchmarking and developing perceptual similarity metrics. With UniSim-Bench, we combine several perceptual tasks (Sec. 3.1) into a cohesive benchmark (Sec. 3.2): while diverse, we consider these tasks as specific instances of a broader challenge, i.e. capturing the human perception of similarity. We first detail the individual perceptual tasks (Sec. 3.1), then present the structure of our benchmark (Sec. 3.2). Finally, in Sec. 3.3 we develop our multi-task perceptual models, UniSim.

### 3.1 Multi-modal perceptual similarity tasks

The following tasks, visualized in Fig. 1 and Fig. 2, are the basis of our benchmark (further details in App. B).

**Image-to-Image Similarity ( Img-2AFC ).** Each data point consists of a triplet $(x_{\text{ref}}, x_1, x_2)$, and one has to decide which of two images $x_1, x_2$ is more similar to the reference image $x_{\text{ref}}$. The BAPPS (Zhang et al., 2018) (used to tune the LPIPS metric) and PIEAPP (Prashnani et al., 2018) datasets contain images perturbed with different corruptions and compare their similarity to the original images. Conversely, NIGHTS (Fu et al., 2023) includes high-resolution synthetic images and aims at capturing more high-level similarities (pose, perspective, number of items, etc.). The labels describe the human preference between the alternative images.

**Image-to-Text Alignment ( IT-2AFC ).** Perceptual metrics are used to assess how well synthetic images generated by text-to-image models align with their prompts. Each sample consists of a prompt $t_{\text{ref}}$, a set of generated images, and human annotations—provided as scores or rankings—that evaluate the images' alignment to the prompt. To standardize the task structure, we randomly sample two images $x_1, x_2$ and form the triplet $(t_{\text{ref}}, x_1, x_2)$, and the label corresponds to the higher-scoring or higher-ranked image. We use the IMAGEREWARD (Xu et al., 2023), HPDv2 (Wu et al., 2023b), AGIQA-3K (Li et al., 2023a) datasets. Also, we adapt the MAGICBRUSH (Zhang et al., 2024a) and HQ-EDIT (Hui et al., 2024) datasets from instruction-guided image editing to the IT-2AFC task since they include annotated source and target images along with textual instructions.

**Text-to-Image Alignment ( Text-2AFC ).** Assessing the quality and specificity of generated captions for a given image is essential for ensuring accurate and meaningful text generation. Thus, we incorporate the Text-2AFC task, which can be seen as the reverse of IT-2AFC, where the goal is to select the text $t_1$ or $t_2$ that better describes the reference image $x_{\text{ref}}$. We use three datasets: POLARIS (Wada et al., 2024), $\mathcal{CD}$-COCO (Bianco et al., 2023) (based on MS-COCO (Lin et al., 2014)) and HQ-EDIT (Hui et al., 2024).

**Image Quality Assessment ( IQA ).** In this well-established task, one has to determine which of two images $x_1, x_2$ has higher quality. While some works focus on no-reference quality assessment (i.e., assigning an absolute score), we limit our evaluation to pairwise comparisons. The KADID-10K dataset (Lin et al., 2019) contains images artificially corrupted with varying levels of severity, while KONIQ-10K (Hosu et al., 2020) includes images with authentic distortion. Moreover, we again use PIEAPP (Prashnani et al., 2018) and AGIQA-3K (Li et al., 2023a), which also provide quality scores. Finally, we include PIPAL (Jinjin et al., 2020), comprising high-quality reference images subjected to 116 types of distortions.

**Perceptual Attributes Assessment ( PAA ).** Here, we evaluate specific visual characteristics (attributes) of the image that directly affect human perception and contribute to overall visual quality. The perceptual attributes considered are *brightness*, *colorfulness*, *contrast*, and *sharpness*. For an image pair

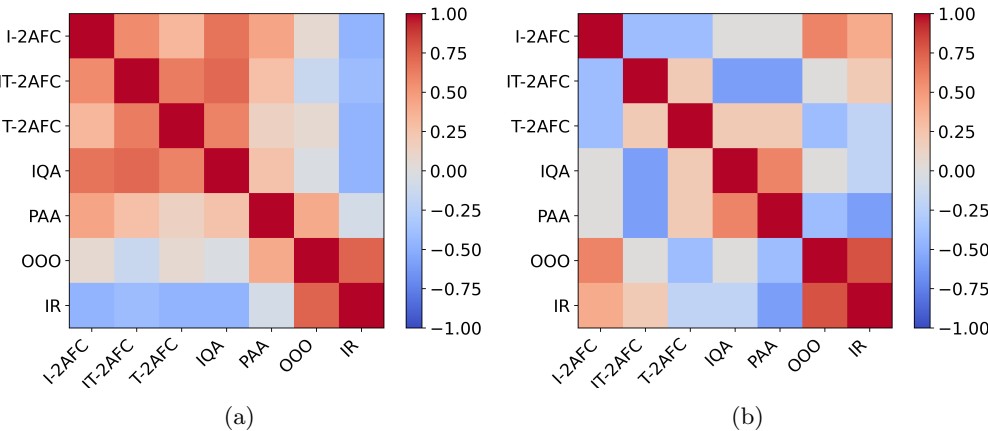

(a)                                                                 (b)

Figure 3: **Correlation maps of model performance across tasks. (a)** General-purpose models exhibit good correlations across core 2AFC tasks. **(b)** With specialized models correlation is weak or negative suggesting overfitting to their narrow tasks.

$(\boldsymbol{x}_1, \boldsymbol{x}_2)$, the task is to choose which one better exhibits the given attribute (e.g., which is brighter). We use the KONIQ-10K (Hosu et al., 2020) dataset for all attributes and additionally leverage the SICE (Cai et al., 2018) dataset for brightness.

**Odd-One-Out ( OOO ).** Given a triplet of images $(\boldsymbol{x}_1, \boldsymbol{x}_2, \boldsymbol{x}_3)$, the task consists of finding the one that does not belong with the others—that is, the most dissimilar image. We use the dataset derived by (Muttenthaler et al., 2023) from the coarse CIFAR-100 classes, named CIFAR-100-OOO. Moreover, we follow a similar approach to obtain IMAGENET-OOO. We get 6 macro-classes by merging certain IMAGENET-1k classes, which yields semantically distinct classes with sufficient intra-class diversity, making the task non-trivial. A triplet consists of two images from the same macro-class and one from a different one, the ground-truth odd one.

**Image-to-Image Retrieval ( IR ).** Perceptual metrics have been used to identify the closest matches to a query image within a database. Unlike the previous tasks, retrieval involves ranking the entire pool of images rather than choosing between 2-3 alternatives. Nevertheless, we consider it a relevant use case for perceptual similarity metrics and include it in our framework. We employ the $\mathcal{R}$OXFORD and $\mathcal{R}$PARIS datasets (Philbin et al., 2008).

### 3.2 UniSim-Bench: an open-ended multi-modal perceptual similarity benchmark

Building on the multi-modal perceptual tasks from Sec. 3.1, we now present our unified framework UniSim-Bench.

**Composition.** We split the tasks from Sec. 3.1 into two groups: the first consists of the ***Core 2AFC Tasks***—Img-2AFC, IT-2AFC, Text-2AFC, and IQA—which form a diverse set of complementary tasks to evaluate different aspects of perceptual similarity. These tasks are well-established with numerous datasets available that can be framed as 2AFC problems. Each task also has multiple specialized models, providing strong baselines for our evaluation. The second group consists of the ***OOD Generalization Tasks***, including PAA, OOO, and IR, which capture more peripheral yet important aspects of perception. These tasks assess how well perceptual metrics trained on core tasks generalize to new unseen perceptual challenges and can be seen as out-of-distribution (OOD) compared to the core tasks. Together, the two splits form UniSim-Bench, which includes 7 tasks and 25 datasets (details in App. B).

**Correlation between tasks.** To better understand the relationship among tasks in UniSim-Bench, we compute Kendall's $\tau$ correlation between the performance of existing perceptual metrics across task pairs. Fig. 3 illustrates the correlation maps among general-purpose models (Fig. 3a) and specialized models fine-tuned for specific tasks (Fig. 3b). We use the same models as in the main experiments in Sec. 4.2. General-

purpose models exhibit positive correlation values across *Core 2AFC Tasks*, which, however, become very weak or even negative for the specialized models. This indicates that fine-tuning on any individual task deteriorates the zero-shot ability of the original features, including between very closely related tasks such as IT-2AFC and Text-2AFC (more on this in Sec. 4.2). Moreover, the *OOD Generalization Tasks* show no correlation with the *Core 2AFC Tasks*, confirming their role of out-of-distribution control tasks. These results highlight the complementarity of our collected tasks and their ability to capture diverse capabilities of perceptual metrics.

**Discussion.** In designing our benchmark, we aimed to capture a wide range of perceptual similarity tasks to enable a comprehensive evaluation of existing automated metrics. While this set, to the best of our knowledge, forms the broadest benchmark currently available for the topic, we consider it an open-ended effort. Future expansions could include additional applications of perceptual similarity metrics and higher-quality datasets for existing tasks. Despite potential limitations, we believe our benchmark provides valuable insights into the shortcomings of current metrics and offers a foundation for the development of more robust metrics across diverse modalities and applications, as explored in the following sections.

### 3.3 UniSim: a family of multi-task perceptual similarity metrics

We now introduce UniSim, our unified multi-task perceptual metric (additional information in App. B).

**UniSim training data.** UniSim is trained on a subset of datasets from the core tasks of UniSim-Bench, as detailed in App. B, while the *OOD Generalization Tasks* are entirely excluded from training. Additionally, certain datasets from the *Core 2AFC Tasks* (i.e., BAPPS, IMAGEREWARD, AGIQA-3K, $\mathcal{CD}$-COCO, KONIQ-10K) are deliberately withheld for evaluating generalization. This approach allows us to assess three types of generalization of perceptual metrics: 1) standard *training-test* set generalization (for datasets included in training), 2) *intra-task* generalization, where the model is tested on unseen datasets from known (core) tasks, 3) *inter-task* generalization, where the model is tested on new, unseen (non-2AFC) tasks. In training on multiple *Core 2AFC Tasks*, one major challenge is unifying datasets with diverse formats. While *Core 2AFC Tasks* vary in structure, we have standardized them into a 2AFC format as explained in Sec. 3. Each data point in the 2AFC structure is a triplet $(z_{\text{ref}}, z_0, z_1)$ consisting of text prompts or images, with a reference item $z_{\text{ref}}$ and two alternatives $z_0, z_1$, as well as a label $y \in \{0, 1\}$ indicating which alternative is more similar to the reference.[1] This setup allows us to frame the problem as binary classification.

**CLIP-based UniSim.** To fine-tune a CLIP model to solve the binary classification problem, we optimize the hinge loss, as in earlier methods (Liu et al., 2021)

$$\mathcal{L}(z_{\text{ref}}, z_0, z_1, y, \phi, \psi) = \max\{0, (2y - 1) \cdot (\text{sim}_{\phi,\psi}(z_{\text{ref}}, z_0) - \text{sim}_{\phi,\psi}(z_{\text{ref}}, z_1)) + \mu\}, \tag{1}$$

where $\text{sim}_{\phi,\psi}$ is the similarity function induced by the CLIP model (with encoders $\phi, \psi$, see Sec. 2.2), and $\mu \geq 0$ a margin to ensure confident predictions. We fine-tune only the image encoder $\phi$, i.e., the text encoder $\psi$ is frozen. We concatenate the datasets belonging to the same task and denote the $i$-th data sample for the $t$-th tasks as $(z_{\text{ref}}^{(t,i)}, z_0^{(t,i)}, z_1^{(t,i)}, y^{(t,i)})$, getting the training objective

$$\min_{\phi} \sum_{t=1}^{4} \sum_{i=1}^{n} \mathcal{L}(z_{\text{ref}}^{(t,i)}, z_0^{(t,i)}, z_1^{(t,i)}, y^{(t,i)}, \phi, \psi) \tag{2}$$

where, in practice, we replace $n$ (the entire dataset) with the batch size used for training. This approach ensures that the number of samples seen is balanced across tasks, regardless of the dataset size. Following (Liu et al., 2021; Croce et al., 2025) we use LoRA (Hu et al., 2022) for efficient fine-tuning while mitigating overfitting. We apply this approach to the CLIP model with ViT-B/32 as vision encoder from the OpenCLIP library (Cherti et al., 2023) and to the original CLIP (ViT-L/14 backbone) (Radford et al., 2021), resulting in two encoder-based UniSim models of different size.

**LMM-based UniSim.** For the LMM-based version of our perceptual metric, we fine-tune the LLaVA-NeXT-0.5B model (Li et al., 2024b), as it has shown advanced capability to handle multi-image inputs and

---

[1] For IQA we use the prompt `"A high quality photo."` as reference to complete the triplet

Table 2: **Evaluation on the *Core 2AFC Tasks* of UniSim-Bench.** We provide a comparative analysis of general-purpose, specialized, and UniSim models on the first section of UniSim-Bench. LMM-based models are distinguished with the ♣ symbol, while models highlighted with color are specialized in individual tasks (e.g., DS is specialized for the Img-2AFC task). Additionally, the datasets used for training each model are indicated as superscripts next to their names. **Observations:** (1) Specialized models generally perform worse than general-purpose models on tasks outside their training domain, highlighting a significant lack of generalization. For example, the HPSv2 model, which is specialized for the IT-2AFC task, performs worse than the baseline (ViT-H/14) on the closely related Text-2AFC task. (2) UniSim ranks as the first or second best across nearly all tasks, demonstrating the feasibility of training a unified multi-modal metric capable of handling diverse and widely-used tasks.

| Models | Img-2AFC | | | | IT-2AFC | | | | | | Text-2AFC | | | | IQA | | | | | | Avg |
|---|---|---|---|---|---|---|---|---|---|---|---|---|---|---|---|---|---|---|---|---|---|
| | NIGHTS [1,†] | BAPPS [†] | PIEAPP [†] | average | IMAGEREWARD [3] | HPDv2 [4,†] | AGIQA-3K | MAGICBRUSH [†] | HQ-EDIT [†] | average | CD-COCO | POLARIS [†] | HQ-EDIT [†] | average | KADID-10K [5,†] | KONIQ-10K [6] | PIEAPP [†] | AGIQA-3K | PIPAL [†] | average | |
| **General-purpose models** | | | | | | | | | | | | | | | | | | | | | |
| **CLIP** ViT-B/32 | 85.1 | 68.6 | 80.2 | **78.0** | 65.8 | 63.3 | 66.1 | 72.4 | 85.2 | **70.6** | 61.4 | 78.9 | 84.6 | **75.0** | 59.8 | 51.8 | 80.5 | 68.3 | 74.4 | **67.0** | 72.6 |
| **CLIP** ViT-L/14 | 81.5 | 64.2 | 76.1 | **73.9** | 63.1 | 65.8 | 62.9 | 78.2 | 84.7 | **70.9** | 75.0 | 82.0 | 83.6 | **80.2** | 84.1 | 69.1 | 90.5 | 77.7 | 88.8 | **82.0** | 76.8 |
| **CLIP** ViT-H/14 | 84.0 | 69.0 | 76.8 | **76.6** | 63.3 | 65.5 | 65.1 | 76.5 | 86.5 | **71.4** | 66.4 | 81.8 | 85.6 | **77.9** | 67.0 | 61.1 | 72.0 | 65.7 | 67.5 | **66.7** | 73.1 |
| **BLIP** ViT-L/14 | 80.8 | 65.0 | 72.1 | **72.6** | 64.1 | 67.0 | 64.5 | 73.3 | 85.4 | **70.9** | 66.3 | 78.9 | 82.6 | **75.9** | 65.1 | 55.2 | 61.0 | 57.0 | 61.9 | **60.0** | 69.9 |
| **SigLIP** SoViT/14 | 82.8 | 66.8 | 78.8 | **76.1** | 63.8 | 69.2 | 65.5 | 75.2 | 79.1 | **70.6** | 66.2 | 82.0 | 76.5 | **74.9** | 57.2 | 55.2 | 62.5 | 59.3 | 61.5 | **59.1** | 70.2 |
| **LLaVA-NeXT**-0.5B♣ | 57.1 | 52.8 | 63.0 | **57.6** | 61.3 | 76.6 | 65.2 | 64.4 | 75.1 | **68.5** | 53.7 | 71.6 | 57.9 | **61.1** | 53.6 | 52.7 | 55.1 | 57.5 | 50.8 | **53.9** | 60.3 |
| **LLaVA-NeXT**-7B♣ | 91.3 | 67.0 | 79.9 | **79.4** | 71.5 | 76.1 | 68.5 | 72.7 | 86.5 | **75.1** | 59.6 | 79.4 | 80.0 | **73.0** | 64.1 | 79.2 | 83.6 | **79.7** | 80.9 | **77.5** | 76.2 |
| **Mantis** Idefics-8B♣ | 89.5 | 63.8 | 75.0 | **76.1** | 71.0 | 73.9 | 68.5 | 75.8 | 84.4 | **74.7** | 64.7 | 77.8 | 83.0 | **75.2** | 58.3 | 76.3 | 65.1 | 79.0 | 74.9 | **70.7** | 74.2 |
| **Qwen2-VL**-7B♣ | 88.0 | 58.5 | 73.2 | **73.2** | 54.6 | 39.0 | 49.7 | 58.0 | 50.0 | **50.3** | 50.3 | 50.4 | 50.0 | **50.2** | 63.4 | 61.7 | 56.1 | 49.0 | 58.1 | **57.7** | 57.8 |
| **InternVL2.5**-8B♣ | 85.4 | 56.2 | 69.2 | **70.3** | 68.0 | 69.2 | 68.1 | 82.0 | 87.3 | **74.9** | 65.6 | 81.5 | 87.9 | **78.3** | 70.0 | 68.7 | 66.9 | 72.3 | 69.6 | **69.5** | 73.3 |
| **InternVL2.5**-26B♣ | 88.3 | 59.6 | 73.2 | **73.7** | 69.2 | 74.0 | 69.3 | 85.0 | 88.4 | **77.2** | 68.5 | 83.3 | 87.5 | **79.8** | 76.5 | 74.9 | 76.9 | 76.3 | 81.9 | **77.3** | 77.0 |
| **InternVL2.5**-38B♣ | 89.3 | 61.2 | 78.2 | **76.2** | 62.4 | 73.8 | 68.2 | 90.0 | 88.5 | **76.6** | 70.40 | 81.50 | 87.6 | **79.8** | 73.4 | 69.7 | 75.9 | 75.4 | 82.0 | **75.3** | 77.0 |
| **Specialized models** | | | | | | | | | | | | | | | | | | | | | |
| **DS**[1] ViT-B/32 | 95.3 | **73.3** | 88.5 | **85.7** | 63.1 | 62.0 | 64.4 | 68.8 | 79.8 | **67.6** | 61.3 | 75.6 | 84.1 | **73.7** | 70.1 | 58.0 | 78.4 | 67.1 | 72.7 | **69.2** | 74.1 |
| **DS**[1] Ensemble | **96.2** | 72.5 | **89.1** | **85.9** | - | - | - | - | - | **-** | - | - | - | **-** | - | - | - | - | - | **-** | - |
| **IR**[3] BLIP | 87.1 | 66.1 | 77.6 | **76.9** | **74.3** | 74.5 | 72.4 | 74.3 | 83.5 | **75.8** | 54.2 | 72.2 | 85.4 | **70.6** | 62.3 | 58.0 | 75.1 | 74.8 | 60.1 | **66.1** | 72.3 |
| **HPSv2**[4] ViT-H/14 | 78.5 | 66.7 | 70.8 | **72.0** | 73.8 | **83.5** | **72.6** | 74.9 | 81.2 | **77.2** | 68.2 | 78.1 | 81.5 | **75.9** | 67.0 | 63.6 | 68.9 | 65.4 | 73.5 | **67.7** | 73.2 |
| **PAC-S** ViT-L/14 | 86.9 | 69.1 | 78.1 | **78.0** | 65.0 | 67.0 | 65.8 | 75.6 | 86.9 | **72.1** | 60.5 | 77.6 | 85.6 | **74.6** | 75.0 | 56.5 | 86.1 | 70.0 | 83.2 | **74.2** | 74.7 |
| **LIQE**[5,6] ViT-B/32 | 77.9 | 68.7 | 76.6 | **74.4** | 61.9 | 67.3 | 64.1 | 59.9 | 78.3 | **66.3** | 63.5 | 78.2 | 81.0 | **74.2** | 92.4 | 87.9 | 98.2 | 76.7 | 86.0 | **88.2** | 75.8 |
| **C2S**♣[5,6] mOwl-2 | - | - | - | **-** | - | - | - | - | - | **-** | - | - | - | **-** | **96.2** | **92.0** | 99.2 | 76.3 | 87.3 | **90.2** | - |
| **DeQA-Score**♣ | - | - | - | **-** | - | - | - | - | - | **-** | - | - | - | **-** | 93.9 | 90.7 | **99.4** | 78.8 | 90.4 | **90.6** | - |
| **Our models**[†] | | | | | | | | | | | | | | | | | | | | | |
| **UniSim** ViT-B/32 | 87.7 | 69.9 | 84.6 | **80.7** | 70.4 | 74.5 | 71.7 | 78.1 | 84.1 | **75.8** | 91.2 | 94.2 | 85.6 | **90.3** | 89.9 | 72.0 | 93.6 | 77.3 | **93.4** | 85.3 | 83.0 |
| **UniSim** ViT-L/14 | 90.7 | 68.1 | 85.0 | **81.3** | 69.4 | 82.3 | 71.3 | **91.8** | 86.0 | **80.2** | 94.2 | 96.1 | 88.3 | **92.9** | 94.7 | 71.8 | 98.9 | 80.2 | 89.2 | 87.0 | **85.3** |
| **UniSim**♣ LL-N-0.5B | 89.8 | 70.0 | 85.3 | 81.7 | 69.2 | 80.7 | 66.7 | 90.8 | **92.7** | 80.0 | 75.4 | 99.9 | 89.2 | 88.2 | 94.3 | 77.6 | 97.0 | 80.6 | 89.8 | 87.9 | 84.4 |

image-text interleaved formats. Moreover, LLaVA-NeXT-0.5B has a relatively small number of parameters, making it significantly more efficient for training and inference—an important factor for real-world deployment. For the training, we leverage the instruction fine-tuning mechanism of LLaVA-NeXT-0.5B, and format our datasets to produce the annotation files compatible with those in (Li et al., 2024b), where the tasks are described as natural language instructions (see Sec. 2.2). A significant challenge in fine-tuning LMMs for 2AFC tasks is that the ground truth consists of a single word representing the model's prediction between two alternatives. To mitigate the risk of overfitting to specific structural patterns 1) we design a variety of templates for both instructions and answers, and 2) we combine the Multi-image (500K) part of M4-Instruct (Li et al., 2024b) dataset with our perceptual dataset (842K) ensuring diversity in the training data. Notably, the datasets we curated for the training can be seamlessly integrated with any instruction-tuning dataset to fine-tune LMMs, enhancing their fine-grained perceptual capabilities.

# 4 Evaluation on UniSim-Bench

Next, we use UniSim-Bench for a comprehensive analysis of general-purpose, specialized, and our UniSim models. After introducing the baselines (Sec. 4.1), we discuss the results on core (Sec. 4.2) and generalization tasks (Sec. 4.3).

## 4.1 Baselines

**General-purpose multi-modal models.** For encoder models, we benchmark the CLIP models with ViT-B/32, (which serves as the baseline for both DreamSim, LIQE, and UniSim-ViT-B/32), ViT-L/14 (baseline for PAC-S and UniSim-ViT-L/14), as well as ViT-H/14 (baseline for the HPS-v2 model). We further test SigLIP SoViT-400m/14 (Alabdulmohsin et al., 2024) (results in appendix), and BLIP-2 (Li et al., 2023b) (with a ViT-L/14 encoder), which is the base model for ImageReward. Among LMMs we include Llava-NeXT-0.5B (Li et al., 2024b) (basis of the LLM-based UniSim), its larger version Llava-NeXT-7B (Li et al., 2024b) and the recent Mantis Idefics2-8B (Jiang et al., 2024) (results in appendix), which are specifically multi-image autoregressive models. Additionally, we include Qwen2-VL-7B (Wang et al., 2024b) and InternVL2.5-8B/26B/38B (Chen et al., 2024), two families of recent LMMs demonstrating strong general visual reasoning capabilities. For LMMs, we further test in-context learning strategies (Brown et al., 2020), however, these approaches fail to enhance the zero-shot performance; see App. C.

**Specialized perceptual metrics.** For Img-2AFC, DreamSim (DS) (Fu et al., 2023) achieves SOTA performance via an ensemble of multiple vision encoders fine-tuned on NIGHTS: since this is not associated with a text-encoder, we primarily compare their single-encoder (ViT-B/32) version. For the IT-2AFC task, we select the ImageReward (IR) model (Xu et al., 2023) and HPSv2 (Wu et al., 2023b): these are trained on the IMAGEREWARD and HPDv2 datasets respectively for evaluating text-to-image generative models. As a metric specialized in Text-2AFC, we report the results of PAC-S (Sarto et al., 2023), designed for image captioning evaluation. Finally, for IQA we report LIQE (Zhang et al., 2023b) as the encoder model and Compare2Score (Wu et al., 2024) and DeQA-Score (You et al., 2025) (fine-tuned from mPLUG-Owl2-8B (Ye et al., 2024)) as generative baseline models. We provide more details on the models in App. B, and the evaluation of additional baselines in App. C.

## 4.2 Evaluation on *Core 2AFC Tasks*

**Setup.** In Table 2 we measure the performance of the perceptual metrics in terms of classification accuracy. Besides the performance on each dataset, we report the performance per task (average of the accuracy on the individual datasets within the same task) in the column with background color, and the overall average (last column), i.e., the mean of the single-task average performance values.

**Intra-task generalization.** Among the three tiers of generalization we evaluate, the standard *training-test* set generalization is typically achieved by all specialized models and UniSim. However, *intra-task* generalization—where models are tested on unseen datasets within their training tasks—poses a significant challenge for most specialized models. While some models (DreamSim, LIQE) exhibit this capability, others fall short. For example, both HPSv2 and ImageReward perform worse than the generalist baselines on HQ-EDIT, highlighting that existing approaches still struggle with intra-task generalization. Conversely, the UniSim models successfully generalize to the intra-tasks datasets and outperform the baseline on the left-out datasets, sometimes of a large margin e.g., on $\mathcal{CD}$-COCO.

**Inter-task generalization.** Table 2 indicates that models specialized for a single perceptual task often suffer performance degradation on tasks outside their training domain (see also Fig. 1). For instance, DreamSim achieves the best performance on Img-2AFC but underperforms compared to CLIP on IT-2AFC and Text-2AFC. This is likely due to overfitting to the narrow perceptual task, and fine-tuning on a vision-only task may adversely impact image-text alignment. Similarly, HPSv2, specialized for IT-2AFC, underperforms compared to the baseline (CLIP with ViT-H/14) on Text-2AFC, highlighting a lack of generalization even across closely related tasks. In contrast, UniSim consistently ranks as the first or second best across

nearly all tasks and achieves the best average performance. It even improves upon the models from which it was fine-tuned across all tasks. Although this performance may be expected given its training on multiple datasets, it demonstrates the feasibility of a unified multi-modal metric that can effectively handle diverse, widely-used tasks.

### 4.3 Evaluation on *OOD Generalization Tasks*

Table 3 reports the results on the *OOD Generalization Tasks* of the models from Table 2 (average accuracy over datasets is shown, detailed results in App. C). The average performance on these unseen tasks (last column) is lower for perceptual models (both specialized and multi-task) compared to the general-purpose baselines. This aligns with the previous observation that training on a subset of perceptual similarity tasks does not improve performance on unseen tasks. However, for perceptual attributes assessment (PAA), specialized models often achieve accuracy close to or slightly exceeding that of the baselines. For example, all UniSim models outperform their baseline models from which they are fine-tuned. We hypothesize that PAA is a near-OOD task, more similar to the training tasks than odd-one-out or retrieval (far-OOD), making multi-task training slightly beneficial in this context. Unlike for the core tasks (Table 2), the performance of LMMs is generally worse than with CLIP models. The accuracy of the LLM-based UniSim drops significantly on OOO, likely due to the task structure which differs from the 2AFC training data. Conversely, CLIP models are unaffected as each input item is encoded independently regardless of the task, demonstrating stronger generalization capabilities than LMMs.

### 4.4 Additional Analyses

**Evaluation across model scales.** We investigate how model performance varies with scale. Within the InternVL2.5 family, increasing the model size from 7B to 26B yields consistent improvements on *Core 2AFC Tasks* (average accuracy rising from 73.0 to 77.0). However, performance plateaus at 38B, with no further gains observed, suggesting that scaling alone is insufficient to address key perceptual challenges. A similar trend emerges on *OOD Generalization Tasks*: while InternVL2.5-26B outperforms the 8B model, the 38B variant exhibits a regression, with PAA accuracy falling below that of 8B. These findings indicate that larger model size does not necessarily confer stronger perceptual capabilities in vision–language models.

Table 3: **Evaluation on the *OOD Generalization Tasks* of UniSim-Bench.** The average performance on these unseen tasks (last column) is lower for both specialized perceptual models and our multi-task models compared to the general-purpose baselines. **Observations:** (1) Perceptual metrics generally underperform general-purpose baselines on out-of-distribution tasks (OOO, IR). (2) Encoder-based models (CLIP) demonstrate stronger OOD generalization compared to LMMs, which tend to overfit to the training task structure.

| Models | PAA | OOO | IR | Avg |
|---|---|---|---|---|
| **General-purpose models** | | | | |
| **CLIP** ViT-B/32 | 70.6 | **71.3** | 43.8 | **61.9** |
| **CLIP** ViT-L/14 | 66.8 | 65.8 | 45.5 | 59.4 |
| **CLIP** ViT-H/14 | 68.2 | 70.3 | **50.2** | 62.9 |
| **SigLIP** SoViT/14 | 68.3 | 67.1 | 52.9 | 62.8 |
| **BLIP** ViT-L/14 | 66.1 | 63.4 | 35.9 | 55.1 |
| **LLaVA NeXT**-0.5B♣ | 63.0 | 33.0 | - | - |
| **LLaVA NeXT**-7B♣ | 67.8 | 60.4 | - | - |
| **Mantis** Idefics-8B♣ | 68.2 | 44.1 | - | - |
| **Qwen2-VL**-7B♣ | 59.1 | 49.7 | - | - |
| **InternVL2.5**-8B♣ | 60.7 | 53.5 | - | - |
| **InternVL2.5**-26B♣ | 61.1 | 62.7 | - | - |
| **InternVL2.5**-38B♣ | 57.8 | 56.4 | - | - |
| **Specialized models** | | | | |
| **DreamSim** ViT-B/32 | 70.7 | 61.4 | 38.0 | 56.6 |
| **DS** Ensemble | - | 65.0 | 43.1 | - |
| **ImageReward** BLIP | 65.1 | 70.2 | 41.7 | 59.0 |
| **HPSv2** ViT-H/14 | 67.9 | 56.4 | 36.4 | 53.6 |
| **PAC-S** ViT-L/14 | 65.8 | 71.2 | 48.0 | 61.6 |
| **LIQE** ViT-B/32 | 71.0 | 60.1 | 18.8 | 49.9 |
| **C2S**♣ mOwl-2 | 61.2 | - | - | - |
| **Our models** | | | | |
| **UniSim** ViT-B/32 | **72.9** | 61.9 | 34.2 | 56.3 |
| **UniSim** ViT-L/14 | 67.6 | 53.7 | 25.1 | 48.8 |
| **UniSim**♣ LL-N-0.5B | 64.8 | 24.2 | - | - |

**Varying the IT-2AFC training data.** We study here the effect of varying the number of datasets used for training UniSim models. In particular, we focus on IT-2AFC, and report in Table 4 the results when fine-tuning CLIP models with various configurations. The default UniSim training uses three datasets (HQ-EDIT, HPDv2, MAGICBRUSH), and we test using either

Table 4: **Varying the IT-2AFC training data.** UniSim is trained on IT-2AFC datasets (HQ-Edit, HPDv2, MagicBrush): we study how using either just one or two influences the intra-task generalization (ImageReward, Agiqa-3k) and performance on Text-2AFC.

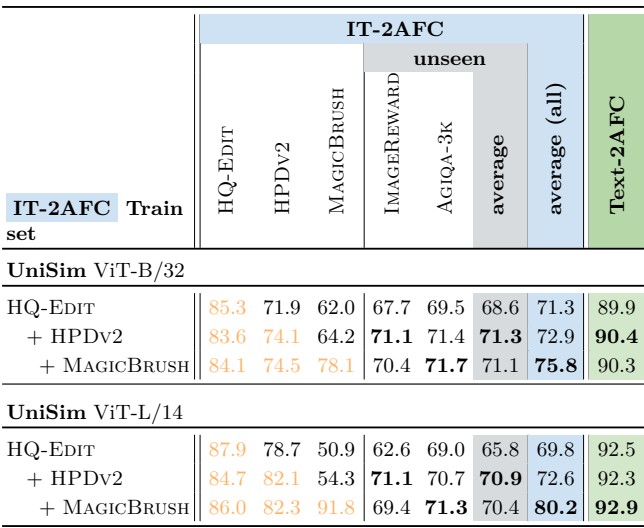

| IT-2AFC Train set | IT-2AFC HQ-Edit | HPDv2 | MagicBrush | unseen ImageReward | Agiqa-3k | average | average (all) | Text-2AFC |
|---|---|---|---|---|---|---|---|---|
| **UniSim** ViT-B/32 | | | | | | | | |
| HQ-Edit | 85.3 | 71.9 | 62.0 | 67.7 | 69.5 | 68.6 | 71.3 | 89.9 |
| + HPDv2 | 83.6 | 74.1 | 64.2 | **71.1** | 71.4 | **71.3** | 72.9 | **90.4** |
| + MagicBrush | 84.1 | 74.5 | 78.1 | 70.4 | **71.7** | 71.1 | **75.8** | 90.3 |
| **UniSim** ViT-L/14 | | | | | | | | |
| HQ-Edit | 87.9 | 78.7 | 50.9 | 62.6 | 69.0 | 65.8 | 69.8 | 92.5 |
| + HPDv2 | 84.7 | 82.1 | 54.3 | **71.1** | 70.7 | **70.9** | 72.6 | 92.3 |
| + MagicBrush | 86.0 | 82.3 | 91.8 | 69.4 | **71.3** | 70.4 | **80.2** | **92.9** |

Figure 4: **From 2AFC to $N$AFC in image-to-text alignment task.** Encoders models better generalize than LMMs to more alternatives in IT-2AFC (HPDv2 dataset).

just one (HQ-Edit) or two (HQ-Edit + HPDv2) of them (the training datasets for the other tasks are unchanged). We find that using two or three datasets (noting that MagicBrush is relatively small, thus has a limited impact) improves intra-task generalization, as observed on ImageReward and Agiqa-3k. Additionally, this setup also enhances performance on a different yet related task, Text-2AFC, indicating that jointly training on multiple perceptual tasks can be mutually beneficial.

**From 2AFC to $N$AFC in Image-to-Text Alignment task.** Next, we analyze the effect of increasing, at test time, the number of alternative images in IT-2AFC (HPDv2 dataset) from 2 to $N$ (we recall the core tasks in UniSim-Bench are 2AFC). Fig. 4 shows the accuracy of CLIP and LLaVA-based UniSim, and the corresponding baselines, for $N = 2, \dots, 8$. The UniSim models outperform their base models: CLIP-based UniSim maintains nearly 50% accuracy at $N = 8$, three times higher than CLIP. Finally, encoder models significantly outperform LMMs, highlighting a current limitation of LMM-based approaches.

## 5 Conclusion

To advance towards a comprehensive multi-task perceptual model, we introduce UniSim-Bench, a benchmark comprising core multi-modal 2AFC perceptual tasks and out-of-distribution generalization tasks. Our evaluation on UniSim-Bench reveals that single-task specialized metrics often underperform general-purpose models (e.g., CLIP) on unseen datasets/tasks. While prior work (Sundaram et al., 2024) suggests that aligning multi-modal models with human perception can benefit certain downstream applications, our experiments reveal a more nuanced picture: perceptual metrics learning often overfits to the training dataset, impairing generalization even to closely related tasks. Furthermore, while the latest perceptual metrics increasingly rely on generative VLMs as the baseline, our evaluations reveal that these models exhibit weaker generalization compared to encoder-based VLMs due to their tendency to overfit to the structure of the training data. These limitations observed in generalization capabilities raise questions about how well these models align with human perception and highlight the need for more robust multi-modal similarity metrics.

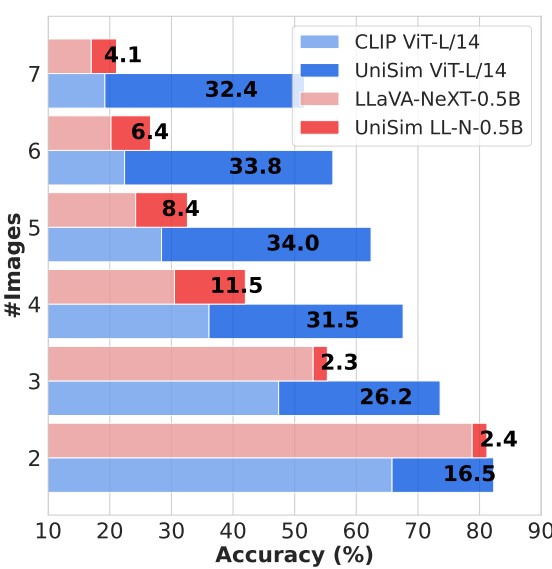

Our multi-task UniSim models represent a step towards this, providing a foundation for future research aimed at comprehensively capturing the human perception of similarity.

## 6 Limitations

In the following, we discuss the limitations of our contributions, distinguishing between the constraints of the benchmark design and the generalization shortcomings of the proposed UniSim models.

**Limitations of UniSim-Bench** As discussed, the collection of tasks and datasets in UniSim-Bench is necessarily limited; for this reason, we formulate it as an open-ended benchmark that can be iteratively enriched by new additions. While the benchmark integrates 7 tasks across 25 datasets to cover a diverse range of uni- and multi-modal challenges, this selection cannot fully capture the complexity of human perception. Furthermore, to standardize the evaluation, we frame the core tasks as Two-Alternative Forced Choice (2AFC) problems. While this structure provides a consistent framework, it inherently restricts evaluation to pairwise comparisons rather than absolute scoring or full ranking, which are standard in tasks such as Image Quality Assessment (IQA) and retrieval.

**Limitations of UniSim Models** Regarding our proposed metrics, we have tested only a small number of base architectures and training configurations, potentially missing more suitable architectures. Our evaluation reveals that while encoder-based models demonstrate superior generalization, generative VLMs (LMMs) tend to overfit to the structure of the training data, resulting in weaker generalization to unseen tasks. Specifically, scaling up the LMM size (e.g., to 7B parameters) led to clear signs of overfitting, where performance on left-out datasets and out-of-distribution (OOD) tasks deteriorated compared to smaller models. Finally, while UniSim improves performance on tasks similar to the training set, generalizing to diverse "Far-OOD" tasks—such as image retrieval and odd-one-out—remains a significant challenge, with performance often lagging behind general-purpose baselines.

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

# A  Extended Related Work

**Image-to-Image Similarity Metrics.**  Recent perceptual metrics have increasingly leveraged deep neural networks to produce data representations, enabling comparisons in the embedding space through measures such as $\ell_p$-norms and cosine similarity (Zhang et al., 2018; Liu et al., 2021; Croce et al., 2025). For image-to-image comparisons, earlier approaches (Zhang et al., 2018; Ghazanfari et al., 2023) utilized the CNN backbones of image classifiers as vision encoders. In contrast, more recent methods (Liu et al., 2021; Croce et al., 2025) exploit modern vision foundation models (Radford et al., 2021; Cherti et al., 2023; Caron et al., 2021), which are trained on vast datasets containing hundreds of millions to billions of samples, to extract highly generalizable visual representations. Additionally, alternative backbones have been explored for visual representation, such as LipSim (Ghazanfari et al., 2024b), which employs Lipschitz networks to enhance robustness against adversarial attacks, and MAE (He et al., 2022), which leverages autoencoders to generate representations.

**Image-to-Text Alignment.**  With the rise of generative models capable of producing images from textual prompts, there has been an increasing demand for robust multi-modal metrics that can effectively evaluate the alignment between the input prompt and the generated image.  CLIP-score (Hessel et al., 2021) and BLIP-score (Li et al., 2023b) are strong candidates for this task, as their vision and text encoders are specifically trained to produce representations that are aligned. However, the primary challenge is that the scores generated by these models are not well aligned with human preference. To address this issue, recent metrics (Xu et al., 2023; Wu et al., 2023b) focus on aligning model evaluations with human preferences. These approaches involve collecting datasets that reflect human judgments by presenting prompts alongside pairs of synthetic images and asking participants to select the image that best aligns with the given prompt. Using this data, ImageReward (Xu et al., 2023) fine-tunes a BLIP model, while HPSv2 (Wu et al., 2023b) fine-tunes a CLIP model, ensuring their outputs are better aligned with human preferences.

**Text-to-Image Alignment.**  Evaluating the correctness and comprehensiveness of generated captions for images is crucial in the evaluation of vision-language models. Similar to image-to-text alignment, the CLIP-score (Hessel et al., 2021) is leveraged for this task. However, the CLIP model is suboptimal for evaluation metrics because its training data lacks the richness and descriptiveness necessary for evaluating generated long captions as investigated by (Sarto et al., 2023).  To address this issue, Sarto et al. (2023) leverage contrastive learning with augmented positive samples to improve the alignment between captions and visual content on the CLIP architecture. Moreover, Polos (Wada et al., 2024) proposes a framework for developing metrics based on human feedback and by leveraging pre-trained CLIP and RoBERTa (Liu et al., 1907) as the encoders. Note that Polos is excluded from our evaluations because it requires an additional text reference, beyond the image-caption pair, to effectively assess the alignment between the caption and the image.

**Image Quality Assessment (IQA).**  With the increasing demand from applications such as super-resolution, denoising, and generative models, the development of advanced IQA methods has gained significant momentum. In this context, foundation models have emerged as the preferred alternative to traditional techniques. Again, vision-language models like CLIP have been effectively employed to compare the visual representations of an image against text prompts describing quality attributes, such as `"A high-quality photo."`. From then new variants of CLIP have been introduced that provide specific setups for training and inference. Recent successful approaches include CLIP-IQA (Wang et al., 2023), which introduces an innovative prompt pairing strategy. This method assesses image quality by utilizing the relative distance between the image and two contrasting prompts: `"Good photo."` and `"Bad photo."`. Moreover, LIQE (Zhang et al., 2023b) proposes a framework for training IQA task along with auxiliary tasks such as scene classification and distortion type identification to enhance the model's generalization.  Additionally, LMMs have been employed for IQA. Notably, Zhu et al. (2024b); Liu et al. (1907); Wu et al. (2024) utilize mPLUG-Owl2(Ye et al., 2024) as their base model, fine-tuning it further on IQA datasets. While mPLUG-Owl2 operates as a single-image LMM, our proposed model harnesses the capabilities of multi-image LMMs, which are better suited for perceptual tasks involving multiple images.

# B    Details on UniSim Framework

In this section, we detail first the various components of the UniSim framework starting with an overview of UniSim-Bench, then the UniSim training process.

## B.1    Perceptual Tasks & Datasets in UniSim-Bench

In the following, each paragraph is dedicated to a specific perceptual task covered in UniSim-Bench and its associated datasets (also summarized in Table 1), and complements the descriptions in Sec. 3.1.

**Image-to-Image Similarity ( Img-2AFC ).** In this task, each data point consists of a triplet $(\boldsymbol{x}_{\text{ref}}, \boldsymbol{x}_1, \boldsymbol{x}_2)$, and one has to decide which of two images $\boldsymbol{x}_1, \boldsymbol{x}_2$ is most similar to the reference image $\boldsymbol{x}_{\text{ref}}$. The BAPPS dataset (Zhang et al., 2018) contains patches of real images perturbed with different corruptions, and compares their similarity to the original images: this was used to tune the LPIPS metric. A similar approach is used to build PIEAPP (Prashnani et al., 2018), where many distortion are applied natural images. Conversely, NIGHTS (Fu et al., 2023) includes high resolution synthetic images, and aims at capturing similarity in terms of pose, perspective, foreground color, number of items, and object shape. All datasets contain labels describing the human preference over the alternative images.

**Image-to-Text Alignment ( IT-2AFC ).** Perceptual metrics are utilized to assess the quality of synthetic images produced by text-to-image generative models (Goodfellow et al., 2020; Ho et al., 2020), evaluating both the overall image quality and the alignment between the provided description and the generated image, ensuring that all relevant details are accurately captured. To achieve this, the IMAGEREWARD (Xu et al., 2023) dataset was curated, comprising six synthetic images for each prompt, with a total of 412 prompts in the test set which are then ranked by experts to capture human preferences for text-to-image generation. For each prompt, we compare the images with highest and lowest rank, to have confident ground-truth labels. Additionally, the HPDv2 dataset (Wu et al., 2023b) was introduced as a large-scale collection aimed at capturing human preferences across a wide variety of image sources. It comprises 798,090 human preference annotations across 433,760 image pairs, making it one of the largest datasets of its kind. The test set samples consist of a prompt, multiple images, and ranks indicating the alignment of each image with the prompt. Following the IT-2AFC setting, two images are randomly selected, and the label is assigned based on their respective rankings. Another dataset utilized in this area is called AGIQA-3K (Li et al., 2023a), designed to evaluate the subjective quality of AI-generated images. It provides subjective scores for two key aspects: perceptual quality, which assesses the overall visual appeal and realism of the images, and text-to-image alignment, which evaluates how well the generated image corresponds to the given textual description. For our benchmark, we first filter out images with low perceptual quality scores. Then, two images are randomly selected and labeled based on the alignment score to form a IT-2AFC sample. The area of instruction-guided image editing features datasets in a structured format, comprising source images, textual instructions, and target images. These datasets naturally align with the IT-2AFC task, as basically, the instruction is a description of the target image. Consequently, we have utilized the MAGICBRUSH (Zhang et al., 2024a) and HQ-EDIT (Hui et al., 2024) from this literature to capitalize on their detailed annotations and structured triplets. HQ-EDIT provides textual descriptions for both the source and target images. Consequently, each sample effectively becomes two distinct samples by utilizing one description at a time and swapping the label accordingly.

**Text-to-Image Alignment ( Text-2AFC ).** The majority of the literature on perceptual metrics has concentrated on evaluating the quality and alignment of synthetic images produced by generative models. However, the reverse task—where an image serves as the input and text is generated as the output—is equally significant. Assessing the quality and specificity of generated captions is essential for ensuring accurate and meaningful text generation. To address this gap, we incorporate the Text-2AFC task, as one of the important tasks for multi-modal perceptual metrics. For this task, we utilize three datasets including POLARIS (Wada et al., 2024), $\mathcal{CD}$-COCO (Bianco et al., 2023) and HQ-EDIT (Hui et al., 2024). The POLARIS dataset consists of 131,020 generated captions and 262,040 reference captions, with human evaluations gathered from 550 participants. Each sample includes an image, a reference caption, and generated captions that received a

score of 0.5 or lower. The $\mathcal{CD}$-Coco (Bianco et al., 2023) benchmark utilizes the MS-COCO (Lin et al., 2014) dataset and generates multiple captions for each image using advanced captioning models and by fusing the top two captions richer, more descriptive captions are created. We utilize 1,000 samples that have human annotations and prune the ones with negative votes and by pairing them with five original captions of MS-COCO data, we create a total of 780 paired samples for evaluations. Finally, the HQ-EDIT dataset, introduced in the previous section, is particularly well-suited for this task as it provides detailed descriptions for both source and target images. Each sample in the Text-2AFC task comprises either a combination of the source image, source description, and target description or the target image paired with the source and target descriptions.

**Image Quality Assessment ( IQA ).** This is an established task where one has to determine which of two images $x_1, x_2$ has higher quality. While there exist works focusing on no-reference quality assessment, i.e., an absolute score, we here restrict our evaluation to pairwise comparison. The KADID-10K dataset (Lin et al., 2019) contains artificially corrupted images with varying levels of severity. Each corrupted image corresponds to a specific reference image. To generate a single sample for IQA, we randomly select an image from the dataset and pair it with another image that represents the next severity level of corruption. Similarly, the KONIQ-10K dataset consists of a pool of images with authentic distortions, from which two images are randomly selected to form a sample Additionally, the PIEAPP dataset can be leveraged by comparing original images with their corresponding corrupted versions. As previously discussed, the AGIQA-3K dataset provides both a perceptual quality score and an image-text alignment score, making it an excellent resource for evaluating the Image Quality Assessment (IQA) by utilizing the the perceptual quality score. Another dataset for IQA is the PIPAL (Jinjin et al., 2020) dataset comprising 29,000 images, including 250 high-quality reference images, each subjected to 116 types of distortions. To ensure reliable subjective quality scores, the dataset includes over 1.13 million human judgments for annotation.

**Perceptual Attributes Assessment ( PAA ).** This task refers to the evaluation of specific visual characteristics or qualities of the image that directly influence how it is perceived by humans. These attributes are subjective and involve measuring various aspects of the image's appearance that contribute to its overall visual quality. More specifically the perceptual attributes included in our work consist of brightness (the perceived level of light or luminance in the image), colorfulness (the intensity or vibrancy of the colors in the image), contrast (the degree of difference between the darkest and lightest parts of the image) and sharpness (the clarity or focus of details in the image). For brightness evaluation, we utilize the SICE (Cai et al., 2018) dataset, while for other attributes, including brightness, we leverage the KONIQ-10K (Hosu et al., 2020) dataset. More specifically, both datasets contain a pool of images with varying levels of the associated perceptual attribute. To create a sample, two images are randomly selected from the pool, and the label is assigned to the image with the higher perceptual attribute level.

**Odd-One-Out ( OOO ).** Given a triplet of images, the task consists in finding the one that does not belong with the others, that is the most dissimilar one. We use the dataset derived by (Muttenthaler et al., 2023) from the coarse CIFAR-100 classes, named CIFAR-100-OOO. Moreover, we follow a similar approach to obtain IMAGENET-OOO: we create 6 macro-classes (`aquatic animals`, `terrestrial animals`, `clothes`, `transportations`, `places`, `musical instruments`) merging a subset of the IMAGENET-1k classes: in this way we get sufficiently semantically separated classes but with enough intra-class diversity so that the tasks is not trivial. Then, for each triplet we sample two images from a macro-class and one from another, which is the ground-truth odd-one-out image. We name this dataset IMAGENET-OOO.

**Image Retrieval ( IR ).** Perceptual metrics have long been employed to identify the closest matches to a query image within a database of images. In this work we employ the revisited versions of Oxford and Paris datasets (Philbin et al., 2008). Both datasets offer three evaluation protocols (easy, medium, hard) to assess performance across varying difficulty levels. $\mathcal{R}$OXFORD contains around 5,000 images in the retrieval pool, while $\mathcal{R}$PARIS includes around 6,000 images, and each use 70 query images. For our evaluations, we report the average accuracy on the medium and hard difficulty levels.

Table 5: **Overview of UniSim training data.** We report the composition of the dataset used for fine-tuning the UniSim models. The number of samples refers to the total contained in the datasets, but might differ from what effectively seen during training by the UniSim models (for example we balance the number of samples from each task while fine-tuning CLIP).

| Task | Dataset | Type | Training samples |
|---|---|---|---|
| **Img-2AFC** | Nights Fu et al. (2023) | Synthetic | 15.9K |
| | PieAPP Prashnani et al. (2018) | Realistic | 50.5K |
| **IT-2AFC** | HPDv2 Wu et al. (2023b) | | 645.1K |
| | MagicBrush Zhang et al. (2024a) | Synthetic | 11.5K |
| | HQ-Edit Hui et al. (2024) | | 100K |
| **Text-2AFC** | Polaris Wada et al. (2024) | Realistic | 245.9K |
| | HQ-Edit Hui et al. (2024) | Synthetic | 100K |
| **IQA** | Kadid-10k Lin et al. (2019) | | 9.1K |
| | PieAPP Prashnani et al. (2018) | Realistic | 50.5K |
| | Pipal Jinjin et al. (2020) | | 73.7K |
| **Total** | | | **1.3M** |

## B.2 UniSim Training

In this section, we present the implementation details of our proposed perceptual metrics, UniSim-CLIP and UniSim-LL-N, which are based on encoder and generative multi-modal models, respectively.

**UniSim-CLIP: Encoder-based Perceptual Metric.** For the training of UniSim-CLIP, we experiment with different versions of CLIP, including ViT-B/32 and ViT-L/14 (336x336 input resolution), which vary in patch size, model size and image resolution. For the training data, the datasets presented in Table 5 are utilized. To ensure a balanced number of samples across tasks, we randomly select 400K samples from each task, resulting in a total of 1.6M samples for training. To ensure consistency, a unified training configuration is employed across all versions, including the use of hinge loss with a margin of 0.05, a batch size of 32, only one epoch with a maximum learning rate of $5 \times 10^{-6}$, a weight decay of 0.35, and a warm-up period of 500 steps, following a cosine learning rate schedule. Moreover, we leverage LoRA (Low-Rank Adaptation) (Hu et al., 2022) (with rank=16, alpha=32, and dropout=0.2) as employed in the previous works (Liu et al., 2021; Croce et al., 2025) to enable efficient fine-tuning while mitigating overfitting.

**UniSim-LL-N: LMM-based Perceptual Metric.** To train UniSim-LL-N, we choose the LLaVA-NeXT (Li et al., 2024b) as the base model leveraging its advanced capability to handle multi-image inputs and image-text interleaved formats. LLaVA-NeXT, which relies on SigLIP-400M/14 vision encoder and the Qwen-1.5 language model (LLM), has two versions with different sizes: LLaVA-NeXT-0.5B and LLaVA-NeXT-7B. One significant challenge in fine-tuning LMMs for perceptual tasks is that the ground truth typically consists of a single word representing the model's prediction between two alternatives. For training, we initially utilized our unified perceptual dataset, see Table 5, annotated with four distinct tasks: Img-2AFC (120K samples), IT-2AFC (300K samples), Text-2AFC (300K samples), and IQA (120K samples). It is important to note that the number of samples for each task varies based on the complexity of the respective task. Additionally, we create another training dataset for UniSim-LL-N, which incorporates the multi-image section of the M4-Instruct dataset (Li et al., 2024b), consisting of 500K samples, added to the UniSim data. We discuss in App. C how including this additional non-perceptual data for training helps improving generalization. For the LLaVA-NeXT-0.5B the entire model, including the vision tower, adapter, and language model, is fine-tuned with learning rate $10^{-5}$ for all components, except $2 \times 10^{-6}$ for the vision tower. While for LLaVA-NeXT-7B the adapter, and language model, are fine-tuned with $2 \times 10^{-6}$ learning

Table 6: **Full evaluation on the *Core 2AFC Tasks* of UniSim-Bench.** We complement the results of Table 2 with additional metrics. LMM-based models are distinguished with the ♣ symbol, while models highlighted with color are specialized in individual tasks (e.g., DS is specialized for the Img-2AFC task). For LLaVA-based UniSim, $v_1$ is trained on perceptual data only (while the default version also uses the multi-image portion of LLaVA-NeXT data, see App. C). The datasets used for training each model are indicated as superscripts next to their names. Moreover, the values in subscripts denote the 95% confidence interval

| Models | Img-2AFC | | | | IT-2AFC | | | | | | Text-2AFC | | | | IQA | | | | | | Avg |
|---|---|---|---|---|---|---|---|---|---|---|---|---|---|---|---|---|---|---|---|---|---|
| | NIGHTS [1,†] | BAPPS | PIEAPP [†] | average | IMAGEREWARD [3] | HPDv2 [4,†] | AGIQA-3K | MAGICBRUSH [†] | HQ-EDIT [†] | average | CD-COCO | POLARIS [†] | HQ-EDIT [†] | average | KADID-10K [5,†] | KONIQ-10K [6] | PIEAPP [†] | AGIQA-3K | PIPAL [†] | average | |
| **General-purpose models** | | | | | | | | | | | | | | | | | | | | | |
| **CLIP** ViT-B/32 | 85.1$^{\pm1.6}$ | 68.6$^{\pm1.3}$ | 80.2$^{\pm1.4}$ | 78.0 | 65.8$^{\pm4.6}$ | 63.3$^{\pm1.3}$ | 66.1$^{\pm1.3}$ | 72.4$^{\pm3.3}$ | 85.2$^{\pm1.6}$ | 70.6 | 61.4$^{\pm3.4}$ | 78.9$^{\pm1.1}$ | 84.6$^{\pm1.6}$ | 75.0 | 59.8$^{\pm1.4}$ | 51.8$^{\pm1.4}$ | 80.5$^{\pm1.1}$ | 68.3$^{\pm1.3}$ | 74.4$^{\pm1.56}$ | 67.0 | 72.6 |
| **CLIP** ViT-L/14 | 81.5$^{\pm1.6}$ | 64.2$^{\pm1.3}$ | 76.1$^{\pm1.5}$ | 73.9 | 63.1$^{\pm4.7}$ | 65.8$^{\pm1.3}$ | 62.9$^{\pm1.3}$ | 78.2$^{\pm3.1}$ | 84.7$^{\pm1.6}$ | 70.9 | 75.0$^{\pm3.0}$ | 82.0$^{\pm1.1}$ | 83.6$^{\pm1.6}$ | 80.2 | 84.1$^{\pm1.0}$ | 69.1$^{\pm1.3}$ | 90.5$^{\pm0.8}$ | 77.7$^{\pm1.4}$ | 88.8$^{\pm1.12}$ | 82.0 | 76.8 |
| **CLIP** ViT-H/14 | 84.0$^{\pm1.7}$ | 69.0$^{\pm1.3}$ | 76.8$^{\pm1.4}$ | 76.6 | 63.3$^{\pm4.7}$ | 65.5$^{\pm1.3}$ | 65.1$^{\pm1.3}$ | 76.5$^{\pm3.2}$ | 86.5$^{\pm1.5}$ | 71.4 | 66.4$^{\pm3.3}$ | 81.8$^{\pm1.1}$ | 85.6$^{\pm1.5}$ | 77.9 | 67.0$^{\pm1.3}$ | 61.1$^{\pm1.4}$ | 72.0$^{\pm1.2}$ | 65.7$^{\pm1.3}$ | 67.5$^{\pm1.67}$ | 66.7 | 73.1 |
| **BLIP** ViT-L/14 | 80.8$^{\pm1.8}$ | 65.0$^{\pm1.3}$ | 72.1$^{\pm1.5}$ | 72.6 | 64.1$^{\pm4.6}$ | 67.0$^{\pm1.3}$ | 64.5$^{\pm1.3}$ | 73.3$^{\pm3.3}$ | 85.4$^{\pm1.5}$ | 70.9 | 66.3$^{\pm3.3}$ | 78.9$^{\pm1.1}$ | 82.6$^{\pm1.7}$ | 75.9 | 65.1$^{\pm1.3}$ | 55.2$^{\pm1.4}$ | 61.0$^{\pm1.4}$ | 57.0$^{\pm1.4}$ | 61.9$^{\pm1.7}$ | 60.0 | 69.9 |
| **SigLIP** SoViT/14 | 82.8$^{\pm1.7}$ | 66.8$^{\pm1.3}$ | 78.8$^{\pm1.4}$ | 76.1 | 63.8$^{\pm4.6}$ | 69.2$^{\pm1.3}$ | 65.5$^{\pm1.3}$ | 75.2$^{\pm3.2}$ | 79.1$^{\pm1.8}$ | 70.6 | 66.2$^{\pm3.3}$ | 82.0$^{\pm1.1}$ | 76.5$^{\pm1.9}$ | 74.9 | 57.2$^{\pm1.4}$ | 55.2$^{\pm1.4}$ | 62.5$^{\pm1.3}$ | 59.3$^{\pm1.4}$ | 61.5$^{\pm1.7}$ | 59.1 | 70.2 |
| **LLaVA-NeXT**-0.5B♣ | 57.1$^{\pm2.3}$ | 52.8$^{\pm1.4}$ | 63.0$^{\pm1.6}$ | 57.6 | 61.3$^{\pm4.7}$ | 76.6$^{\pm1.2}$ | 65.2$^{\pm1.3}$ | 64.4$^{\pm3.6}$ | 75.1$^{\pm1.9}$ | 68.5 | 53.7$^{\pm3.5}$ | 71.6$^{\pm1.2}$ | 57.9$^{\pm2.2}$ | 61.1 | 53.6$^{\pm1.4}$ | 52.7$^{\pm1.4}$ | 55.1$^{\pm1.4}$ | 57.5$^{\pm1.4}$ | 50.8$^{\pm1.8}$ | 53.9 | 60.3 |
| **LLaVA-NeXT**-7B♣ | 91.3$^{\pm1.3}$ | 67.0$^{\pm1.3}$ | 79.9$^{\pm1.4}$ | 79.4 | 71.5$^{\pm4.4}$ | 76.1$^{\pm1.2}$ | 68.5$^{\pm1.3}$ | 72.7$^{\pm3.3}$ | 86.5$^{\pm1.5}$ | 75.1 | 59.6$^{\pm3.4}$ | 79.4$^{\pm1.1}$ | 80.0$^{\pm1.4}$ | 73.0 | 64.1$^{\pm1.3}$ | 79.2$^{\pm1.1}$ | 83.6$^{\pm1.0}$ | 79.7$^{\pm1.1}$ | 80.9$^{\pm1.4}$ | 77.5 | 76.2 |
| **Mantis** Idefics-8B♣ | 89.5$^{\pm1.4}$ | 63.8$^{\pm1.3}$ | 75.0$^{\pm1.5}$ | 76.1 | 71.0$^{\pm4.4}$ | 73.9$^{\pm1.2}$ | 68.5$^{\pm1.3}$ | 75.8$^{\pm3.2}$ | 84.4$^{\pm1.6}$ | 74.7 | 64.7$^{\pm3.4}$ | 77.8$^{\pm1.2}$ | 83.0$^{\pm1.6}$ | 75.2 | 58.3$^{\pm1.4}$ | 76.3$^{\pm1.2}$ | 65.1$^{\pm1.3}$ | 79.0$^{\pm1.1}$ | 74.9$^{\pm1.5}$ | 70.7 | 74.2 |
| **Qwen2-VL**-7B♣ | 88.0$^{\pm1.5}$ | 58.5$^{\pm1.4}$ | 73.2$^{\pm1.5}$ | 73.2 | 54.6$^{\pm1.4}$ | 39.0$^{\pm1.4}$ | 49.7$^{\pm3.7}$ | 58.0$^{\pm2.2}$ | 50.0$^{\pm3.5}$ | 50.3 | 50.3$^{\pm1.4}$ | 50.4$^{\pm2.2}$ | 50.0$^{\pm1.3}$ | 50.2 | 63.4$^{\pm1.3}$ | 61.7$^{\pm1.3}$ | 56.1$^{\pm1.4}$ | 49.0$^{\pm1.4}$ | 58.1$^{\pm1.8}$ | 57.7 | 57.8 |
| **InternVL2.5**-8B♣ | 85.4$^{\pm1.6}$ | 56.2$^{\pm1.4}$ | 69.2$^{\pm1.6}$ | 70.3 | 68.0$^{\pm4.5}$ | 69.2$^{\pm1.3}$ | 68.1$^{\pm1.3}$ | 82.0$^{\pm2.9}$ | 87.3$^{\pm1.5}$ | 74.9 | 65.6$^{\pm3.3}$ | 81.5$^{\pm1.1}$ | 87.9$^{\pm1.4}$ | 78.3 | 70.0$^{\pm1.3}$ | 68.7$^{\pm1.3}$ | 66.9$^{\pm1.3}$ | 72.3$^{\pm1.2}$ | 69.6$^{\pm1.5}$ | 69.5 | 73.3 |
| **InternVL2.5**-26B♣ | 88.3$^{\pm1.5}$ | 59.6$^{\pm1.4}$ | 73.2$^{\pm1.5}$ | 73.7 | 69.2$^{\pm4.5}$ | 74.0$^{\pm1.2}$ | 69.30$^{\pm1.3}$ | 85.0$^{\pm2.7}$ | 88.4$^{\pm1.4}$ | 77.2 | 68.5$^{\pm3.3}$ | 83.3$^{\pm1.0}$ | 87.5$^{\pm1.4}$ | 79.8 | 76.5$^{\pm1.2}$ | 74.9$^{\pm1.2}$ | 76.9$^{\pm1.2}$ | 76.3$^{\pm1.3}$ | 81.9$^{\pm1.4}$ | 77.3 | 77.0 |
| **InternVL2.5**-38B♣ | 89.3$^{\pm1.4}$ | 61.2$^{\pm1.4}$ | 78.2$^{\pm1.5}$ | 76.2 | 62.4$^{\pm4.7}$ | 73.8$^{\pm1.3}$ | 68.2$^{\pm1.3}$ | 90.0$^{\pm2.2}$ | 88.5$^{\pm1.4}$ | 76.6 | 70.4$^{\pm3.2}$ | 81.5$^{\pm1.1}$ | 87.6$^{\pm1.4}$ | 79.8 | 73.4$^{\pm1.2}$ | 69.7$^{\pm1.3}$ | 75.9$^{\pm1.2}$ | 75.4$^{\pm1.2}$ | 82.0$^{\pm1.4}$ | 75.3 | 77.0 |
| **Specialized models** | | | | | | | | | | | | | | | | | | | | | |
| **DS**[1] ViT-B/32 | 95.3$^{\pm1.0}$ | 73.3$^{\pm1.2}$ | 88.5$^{\pm1.1}$ | 85.7 | 63.1$^{\pm4.7}$ | 62.0$^{\pm1.3}$ | 64.4$^{\pm1.3}$ | 68.8$^{\pm3.4}$ | 79.8$^{\pm1.8}$ | 67.6 | 61.3$^{\pm3.4}$ | 75.6$^{\pm1.2}$ | 84.1$^{\pm1.6}$ | 73.7 | 70.1$^{\pm1.4}$ | 58.0$^{\pm1.4}$ | 78.4$^{\pm1.1}$ | 67.1$^{\pm1.3}$ | 72.7$^{\pm1.6}$ | 69.2 | 74.1 |
| **DS**[1] Ensemble | 96.2$^{\pm0.9}$ | 72.5$^{\pm1.2}$ | 89.1$^{\pm1.1}$ | 85.9 | - | - | - | - | - | - | - | - | - | - | - | - | - | - | - | - | - |
| **IR**[3] BLIP | 87.1$^{\pm1.5}$ | 66.1$^{\pm1.3}$ | 77.6$^{\pm1.4}$ | 76.9 | 74.3$^{\pm4.2}$ | 74.5$^{\pm1.2}$ | 72.4$^{\pm3.3}$ | 74.3$^{\pm1.6}$ | 83.5$^{\pm3.5}$ | 75.8 | 54.2$^{\pm1.2}$ | 72.2$^{\pm1.5}$ | 85.4$^{\pm1.3}$ | 70.6 | 62.3$^{\pm1.3}$ | 58.0$^{\pm1.4}$ | 75.1$^{\pm1.2}$ | 74.8$^{\pm1.2}$ | 60.1$^{\pm1.7}$ | 66.1 | 72.3 |
| **HPSv2**[4] ViT-H/14 | 78.5$^{\pm1.9}$ | 66.7$^{\pm1.3}$ | 70.8$^{\pm1.5}$ | 72.0 | 73.8$^{\pm4.3}$ | 83.5$^{\pm1.0}$ | 72.6$^{\pm1.2}$ | 74.9$^{\pm3.2}$ | 81.2$^{\pm1.7}$ | 77.2 | 68.2$^{\pm3.3}$ | 78.1$^{\pm1.1}$ | 81.5$^{\pm1.7}$ | 75.9 | 67.0$^{\pm1.3}$ | 63.6$^{\pm1.3}$ | 68.9$^{\pm1.3}$ | 65.4$^{\pm1.3}$ | 73.5$^{\pm1.6}$ | 67.7 | 73.2 |
| **PAC-S** ViT-L/14 | 86.9$^{\pm1.5}$ | 69.1$^{\pm1.3}$ | 78.1$^{\pm1.4}$ | 78.0 | 65.0$^{\pm4.6}$ | 67.0$^{\pm1.3}$ | 65.8$^{\pm1.3}$ | 75.6$^{\pm3.2}$ | 86.9$^{\pm1.5}$ | 72.1 | 60.5$^{\pm3.4}$ | 77.6$^{\pm1.2}$ | 85.6$^{\pm1.5}$ | 74.6 | 75.0$^{\pm1.2}$ | 56.5$^{\pm1.4}$ | 86.1$^{\pm1.0}$ | 70.0$^{\pm1.3}$ | 83.2$^{\pm1.3}$ | 74.2 | 74.7 |
| **LIQE**[5,6] ViT-B/32 | 77.9$^{\pm1.9}$ | 68.7$^{\pm1.3}$ | 76.6$^{\pm1.4}$ | 74.4 | 61.9$^{\pm4.7}$ | 67.3$^{\pm1.3}$ | 64.1$^{\pm1.3}$ | 59.9$^{\pm3.6}$ | 78.3$^{\pm1.8}$ | 66.3 | 63.5$^{\pm3.4}$ | 78.2$^{\pm1.1}$ | 81.0$^{\pm1.7}$ | 74.2 | 92.4$^{\pm0.7}$ | 87.9$^{\pm0.9}$ | 98.2$^{\pm0.4}$ | 76.7$^{\pm1.2}$ | 86.0$^{\pm1.2}$ | 88.2 | 75.8 |
| **C2S**♣[5,6] mOwl-2 | - | - | - | - | - | - | - | - | - | - | - | - | - | - | 96.2$^{\pm0.5}$ | 92.0$^{\pm0.8}$ | 99.2$^{\pm0.2}$ | 76.3$^{\pm1.2}$ | 87.3$^{\pm1.2}$ | 90.2 | - |
| **DeQA**♣[5,6] mOwl-2 | - | - | - | - | - | - | - | - | - | - | - | - | - | - | 93.9$^{\pm0.7}$ | 90.7$^{\pm0.8}$ | 99.4$^{\pm0.2}$ | 78.8$^{\pm1.1}$ | 90.4$^{\pm1.0}$ | 90.6 | - |
| **Our models**[†] | | | | | | | | | | | | | | | | | | | | | |
| **UniSim** ViT-B/32 | 87.7$^{\pm1.5}$ | 69.9$^{\pm1.3}$ | 84.6$^{\pm1.2}$ | 80.7 | 70.4$^{\pm4.4}$ | 74.5$^{\pm1.2}$ | 71.7$^{\pm1.2}$ | 78.1$^{\pm3.1}$ | 84.1$^{\pm1.6}$ | 75.8 | 91.2$^{\pm2.0}$ | 94.2$^{\pm0.6}$ | 85.6$^{\pm1.5}$ | 90.3 | 89.9$^{\pm0.8}$ | 72.0$^{\pm1.2}$ | 93.6$^{\pm0.7}$ | 77.3$^{\pm1.2}$ | 93.4$^{\pm0.9}$ | 85.3 | 83.0 |
| **UniSim** ViT-L/14 | 90.7$^{\pm1.3}$ | 68.1$^{\pm1.3}$ | 85.0$^{\pm1.2}$ | 81.3 | 69.4$^{\pm4.4}$ | 82.3$^{\pm1.1}$ | 71.3$^{\pm1.3}$ | 91.8$^{\pm2.0}$ | 86.0$^{\pm1.5}$ | 80.2 | 94.2$^{\pm1.7}$ | 96.1$^{\pm0.5}$ | 88.3$^{\pm1.4}$ | 92.9 | 94.7$^{\pm0.6}$ | 71.8$^{\pm1.2}$ | 98.9$^{\pm0.3}$ | 80.2$^{\pm1.1}$ | 89.2$^{\pm1.1}$ | 87.0 | 85.3 |
| **UniSim**♣ LL-N-0.5B | 89.8$^{\pm1.4}$ | 70.0$^{\pm1.4}$ | 85.3$^{\pm1.2}$ | 81.7 | 69.2$^{\pm4.5}$ | 80.7$^{\pm1.1}$ | 66.7$^{\pm1.3}$ | 90.8$^{\pm2.2}$ | 92.7$^{\pm1.1}$ | 80.0 | 75.4$^{\pm3.0}$ | 99.9$^{\pm0.1}$ | 89.2$^{\pm1.4}$ | 88.2 | 94.3$^{\pm0.6}$ | 77.6$^{\pm1.2}$ | 97.0$^{\pm0.5}$ | 80.6$^{\pm1.1}$ | 89.8$^{\pm1.1}$ | 87.9 | 84.4 |
| **UniSim**♣,$v_1$ LL-N-0.5B | 91.7$^{\pm1.3}$ | 68.4$^{\pm1.3}$ | 85.3$^{\pm1.2}$ | 81.8 | 72.8$^{\pm4.3}$ | 77.7$^{\pm1.2}$ | 65.8$^{\pm1.3}$ | 96.2$^{\pm1.4}$ | 91.2$^{\pm1.2}$ | 80.7 | 74.4$^{\pm3.1}$ | 99.8$^{\pm0.1}$ | 89.0$^{\pm1.4}$ | 87.7 | 94.9$^{\pm0.6}$ | 70.3$^{\pm1.3}$ | 97.7$^{\pm0.4}$ | 79.6$^{\pm1.1}$ | 89.7$^{\pm1.1}$ | 86.4 | 84.2 |
| **UniSim**♣,$v_1$ LL-N-7B | 92.7$^{\pm1.2}$ | 67.6$^{\pm1.3}$ | 86.6$^{\pm1.2}$ | 82.3 | 60.2$^{\pm4.7}$ | 72.6$^{\pm1.1,2}$ | 65.2$^{\pm1.3}$ | 97.7$^{\pm1.1}$ | 91.7$^{\pm1.2}$ | 77.5 | 71.2$^{\pm3.2}$ | 99.9$^{\pm0.1}$ | 90.7$^{\pm1.2}$ | 87.3 | 93.4$^{\pm0.7}$ | 73.9$^{\pm1.2}$ | 96.6$^{\pm0.5}$ | 81.2$^{\pm1.1}$ | 89.9$^{\pm1.1}$ | 87.0 | 83.5 |

Table 7: **Instructions employed during inference for each perceptual task.** We detail the prompt used for evaluating the LMMs on the various perceptual tasks.

| Tasks | Instruction |
|---|---|
| Img-2AFC | Answer the following multiple-choice question:\nHere are three images: <image><image><image>. \n If image 1 is the reference image, which image of the other two is more similar to the reference image? \n Options: \n(A) Image 2 \n(B) Image 3 |
| IT-2AFC | Answer the following question:\nHere are two images: <image><image>, and here is the reference caption: {prompt}. which of the two images is more aligned to the reference caption?\n Options:\n(A) Image 1 \n(B) Image 2 |
| Text-2AFC | Answer the following multiple-choice question:\nGiven the reference image: <image> and two captions, caption 1: {caption1}, caption 2: {caption2} \n which caption has a better alignment with the reference image? \nOptions:\n(A) Caption 1\n(B) Caption 2 |
| IQA | Answer the following multiple-choice question:\nGiven two images: <image><image> which image has a better quality? \nOptions:\n(A) Image 1\n(B) Image 2 |
| PAA | Answer the following multiple-choice question:\nGiven two images: <image><image> which image is more {perceptual attribute}? \nOptions:\n(A) Image 1 \n(B) Image 2 |
| OOO | Answer the following multiple-choice question:\nHere are three images: <image><image><image>, Which one (A, B, C) is the odd-one-out of the group?\n Options:\n(A) Image 1\n(B) Image 2\n(C) Image 3' |

rate to avoid overfitting to the training data. Weight decay is disabled, and a warm-up ratio of 0.03 of the total training steps is applied. The training is performed for a single epoch, following the standard practice for training LMMs.

# C Additional Experiments

In this section, we begin by providing details on the evaluation setup. Next, we present the complete versions of Tables 2 and 3, including the detailed results over datasets and models omitted in the main part. Finally, we discuss the variations in the UniSim-LL-N models, focusing on differences in their size and training data.

**Evaluation setup.**  While evaluating encoder-based perceptual metrics on IQA, we test two approaches with the encoder models: first, a naive approach computes the alignment between the prompt `"A high quality photo."` (i.e, the reference) and the two alternative images. Second, we apply the CLIP-IQA technique from (Wang et al., 2023), where for each image one measures the similarity to two opposite prompts (`"Good photo."`, `"Bad photo."`), and obtains a quality score as the similarity to the first prompt after softmax normalization. The image with higher quality score is then chosen. For each model we test both approaches and report the results of the one which performs best on average on the task. Finally, we use the same two approaches for PAA, again reporting the best-performing one for the task. For evaluating the LMM-based models, we use specific instructions tailored to each perceptual task. These instructions are detailed in Table 7.

**Complete evaluation.**  Table 6 presents a comprehensive evaluation of perceptual metrics using UniSim-Bench. The table includes SigLIP 400m (Zhai et al., 2023), a variation of CLIP where the softmax function is replaced with a sigmoid function. Additionally, it features DreamSim Ensemble, which integrates DINO (Caron et al., 2021), OpenCLIP (Cherti et al., 2023), and the CLIP model for enhanced performance. Table 8 provides a detailed evaluation of each dataset within *OOD Generalization Tasks*, offering a comprehensive overview of the strengths and weaknesses of each model.

**Other analyses.**  As previously mentioned, two versions of LLaVA-NeXT (0.5B and 7B) are used to train the UniSim-LL-N models. A comparison of these versions (trained only on UniSim training data) is presented in Tables 6 and 8 (marked as $v_1$). Notably, UniSim-LL-N-7B exhibits clear signs of overfitting, performing worse than its baseline on the left-out datasets in *Core 2AFC Tasks* and on most datasets in *OOD Generalization Tasks*. In contrast, UniSim-LL-N-0.5B demonstrates better generalization.

Moreover, we see that UniSim-LL-N-0.5B, trained on both the UniSim training data and a subset of the LLaVA-NeXT data, achieves better generalization performance than UniSim-LL-N-0.5B$^{v_1}$, trained only on the UniSim data (see Table 8). We hypothesize that such additional data reduces overfitting to the specific 2AFC data structure.

**In-Context Learning for LMMs.**  In-context learning (ICL) (Brown et al., 2020) is a technique where a model learns to perform tasks by conditioning its predictions on a small set of input-output demonstration examples provided directly in the context rather than updating the model parameters. Fig. 5 illustrates the effect of applying ICL to LLaVA-NeXT-0.5B/7B and UniSim-LL-N-0.5/7B, which are LMMs. The reported accuracy is averaged across four datasets, including one from each of *Core 2AFC Tasks*. Our experiments demonstrate that, regardless of the number of demonstrations provided (ranging from 1 to 3), the use of ICL consistently reduces the accuracy across all evaluated models. This observation aligns with recent findings (Baldassini et al., 2024; Doveh et al., 2024) that highlight the challenges and non-trivial effectiveness of applying ICL in the context of LMMs.

Table 8: **Detailed evaluation on the *OOD Generalization Tasks* of UniSim-Bench.** To complement the results of Table 3, report the performance of the various perceptual metrics on each dataset included in the *OOD Generalization Tasks*, together with the average performance over tasks.

| Models | PAA | | | | | | OOO | | | IR | | | Avg |
|---|---|---|---|---|---|---|---|---|---|---|---|---|---|
| | SICE-bri | KONIQ-10k-bri | KONIQ-10k-col | KONIQ-10k-con | KONIQ-10k-sha | average | IMAGENET-OOO | CIFAR-100-OOO | average | ROXFORD | RPARIS | average | |
| **General-purpose models** | | | | | | | | | | | | | |
| **CLIP ViT-B/32** | 97.1 | 67.0 | 61.5 | 57.7 | 69.5 | 70.6 | 68.2 | **74.3** | **71.3** | 28.1 | 59.6 | 43.8 | 61.9 |
| **CLIP ViT-L/14** | 94.5 | 58.5 | 57.6 | 58.2 | 65.4 | 66.8 | 69.4 | 62.1 | 65.8 | 31.8 | 59.3 | 45.5 | 59.4 |
| **CLIP ViT-H/14** | 96.3 | 66.1 | 57.0 | 61.1 | 60.5 | 68.2 | 66.7 | 73.9 | 70.3 | 36.8 | 63.6 | 50.2 | 62.9 |
| **SigLIP 400m** | 98.0 | 61.4 | 63.1 | 56.7 | 64.0 | 68.6 | 67.2 | 72.3 | 69.8 | **37.1** | **68.7** | **52.9** | **63.7** |
| **BLIP ViT-L/14** | 94.2 | 64.3 | 54.6 | 57.6 | 59.8 | 66.1 | 61.6 | 65.2 | 63.4 | 19.0 | 52.9 | 35.9 | 55.1 |
| **LLaVA NeXT-0.5B**♣ | 87.6 | 55.8 | 63.4 | 51.8 | 56.3 | 63.0 | 33.0 | 33.0 | 33.0 | - | - | - | - |
| **LLaVA NeXT-7B**♣ | 92.7 | 64.6 | 64.4 | 58.2 | 59.3 | 67.8 | 55.3 | 65.5 | 60.4 | - | - | - | - |
| **Mantis Idefics-8b**♣ | 97.0 | 60.7 | 62.7 | 61.0 | 59.7 | 68.2 | 44.0 | 44.1 | 44.1 | - | - | - | - |
| **Qwen2-VL**-7B♣ | 82.0 | 54.4 | 49.9 | 52.8 | 56.6 | 59.1 | 44.3 | 55.1 | 49.7 | - | - | - | - |
| **InternVL2.5**-8B♣ | 72.8 | 60.5 | 55.1 | 55.4 | 59.5 | 60.7 | 48.3 | 58.6 | 53.5 | - | - | - | - |
| **InternVL2.5**-26B♣ | 71.2 | 63.3 | 53.1 | 55.9 | 62.2 | 61.1 | 55.6 | 69.7 | 62.7 | - | - | - | - |
| **InternVL2.5**-38B♣ | 59.0 | 60.7 | 53.7 | 55.7 | 60.1 | 57.8 | 50.3 | 62.5 | 56.4 | - | - | - | - |
| **Specialized models** | | | | | | | | | | | | | |
| **DS**[1] ViT-B/32 | **99.0** | 66.3 | 63.2 | 58.7 | 66.1 | 70.7 | 59.4 | 63.4 | 61.4 | 25.2 | 50.7 | 38.0 | 56.7 |
| **DS**[1] Ensemble | - | - | - | - | - | - | 64.8 | 69.1 | 67.0 | 27.3 | 57.2 | 42.2 | - |
| **IR**[3] ViT-L/14 | 91.4 | 62.2 | 57.0 | 56.2 | 58.8 | 65.1 | 67.6 | 72.7 | 70.2 | 24.2 | 59.3 | 41.7 | 59.0 |
| **HPSv2**[4] ViT-H/14 | 92.9 | 65.0 | 59.1 | **62.9** | 59.7 | 67.9 | 51.1 | 61.7 | 56.4 | 23.0 | 49.9 | 36.4 | 53.6 |
| **PAC-S**[5] ViT-L/14 | 88.8 | 67.5 | 60.0 | 57.7 | 54.8 | 65.8 | **69.2** | 73.1 | 71.2 | 33.7 | 62.2 | 48.0 | 61.6 |
| **C2S**♣[6,7] mOwl-2 | 63.5 | 62.7 | 51.1 | 57.5 | 71.4 | 61.2 | - | - | - | - | - | - | - |
| **LIQE**[6,7] ViT-B/32 | 92.8 | 68.0 | 58.2 | 60.0 | **75.9** | 71.0 | 65.7 | 54.4 | 60.1 | 12.8 | 24.8 | 18.8 | 49.9 |
| **Our models**[†] | | | | | | | | | | | | | |
| **UniSim** ViT-B/32 | 97.8 | **67.9** | **65.4** | 60.0 | 73.2 | **72.9** | 60.1 | 63.6 | 61.9 | 20.0 | 48.4 | 34.2 | 56.3 |
| **UniSim** ViT-L/14 | 95.4 | 62.1 | 60.8 | 59.3 | 60.3 | 67.6 | 49.6 | 57.7 | 53.7 | 15.8 | 34.3 | 25.1 | 48.8 |
| **UniSim**♣ LL-N-0.5B | 73.0 | 62.3 | 64.8 | 60.1 | 63.8 | 64.8 | 23.7 | 24.6 | 24.2 | - | - | - | - |
| **UniSim**♣,$v_1$ LL-N-0.5B | 68.7 | 62.4 | 61.9 | 61.0 | 63.1 | 63.4 | 15.9 | 16.4 | 16.2 | - | - | - | - |
| **UniSim**♣,$v_1$ LL-N-7B | 71.5 | 56.4 | 58.1 | 51.8 | 61.3 | 59.8 | 24.7 | 15.5 | 20.1 | - | - | - | - |

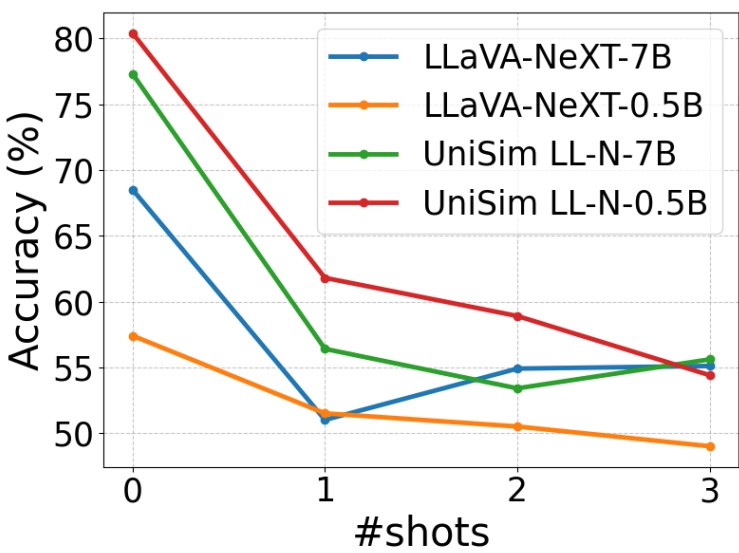

Figure 5: **LMMs with ICL.** We report accuracy (averaged across four datasets, including one from each of *Core 2AFC Tasks*) of LMMs when varying the number of in-context demonstrations. ICL does not help the performance of the perceptual metrics.

