# OpenReview forum: "Towards Unified Benchmark and Models for Multi-Modal Perceptual Metrics"
_TMLR — Rejected by TMLR_

### Review · Reviewer_zVa2 · 2025-11-20

**Summary Of Contributions:**

Summary：

This paper introduces UniSim-Bench, a collection of 25 existing datasets unified into 7 perceptual similarity tasks, and proposes UniSim, a multi-task perceptual metric built by fine-tuning CLIP or LLaVA using LoRA and a standard hinge loss. The authors claim four contributions:

(1) a unified benchmark,
(2) analysis revealing limitations of specialized models,
(3) a multi-task metric (UniSim), and
(4) several analyses on task interactions and generalization.

While the paper is generally well-written and the experiments are extensive, I have several significant concerns regarding novelty, fairness of comparisons, and the validity of the claimed contributions.

Strengths

1. Clear and fluent writing

The paper is very readable, well-organized, and the overall narrative is easy to follow.

2. Large-scale and comprehensive experimentation

The authors conducted evaluations across a wide range of datasets and tasks. The experimental scale is substantial, and the benchmarking effort is commendable.

3. Unifying heterogeneous data into a common 2AFC framework

Despite some issues discussed below, the effort to standardize heterogeneous perceptual datasets into a unified evaluation protocol is valuable and improves reproducibility.

4. Empirical observations on encoder vs. LMM generalization

Although shallow, the observation that encoder-based models generalize better than LMMs is interesting and may be of practical relevance.

Major Weaknesses

1. Contribution #1 (UniSim-Bench): Insufficient originality and lack of deeper analysis

UniSim-Bench is a collection of existing datasets unified under a 2AFC format. This is not inherently a problem, but as a benchmark contribution, the work lacks the level of dataset characterization, distributional analysis, and principled task motivation typically expected in top-tier benchmark papers.

Missing elements include:
a) No statistical analysis of dataset properties (distribution shift, difficulty, noise levels, semantic diversity).
b) No justification for why these specific seven tasks constitute a coherent and complete perceptual similarity space.
c) No analysis of dataset biases or intra-task heterogeneity.

Only a superficial correlation heatmap (Fig. 3), without deeper exploration.
Thus, the benchmark contribution feels more like mechanical aggregation than a principled or insightful benchmark design.

2. Contribution #2 (Limitations of specialized models): Conceptually trivial and empirically unsupported

The claim that single-task specialized metrics lack generalization is well-known in the community and widely understood. To justify this as a contribution, the paper would need fair and rigorous experimental evidence. However, the current evaluation is not symmetry-fair:

a) UniSim is trained on multiple datasets from exactly the same task families used in evaluation.
b) Specialized baselines are trained on single datasets with objectives not aligned to the UniSim-Bench tasks (e.g., DreamSim trained only on Nights, PAC-S trained only for caption evaluation, LIQE only for IQA).
c) Baselines are evaluated far outside their training distribution, while UniSim is evaluated in-distribution.

In this setup, the poor cross-task performance of specialized models is entirely expected and cannot be interpreted as scientific evidence of their limitations. The comparison demonstrates training-data mismatch, not inherent model weakness.

3. Contribution #3 (UniSim model): No methodological novelty

UniSim is not a new model. It is:

a) CLIP or LLaVA
b) fine-tuned via standard LoRA
c) using a standard hinge loss used in LPIPS, DreamSim, and prior perceptual metric works
d) without any architectural changes, new objectives, or algorithmic innovations.

Thus, UniSim’s improvements stem purely from multi-task in-distribution training, not from methodological contributions. This contribution is therefore not convincing.

4. Contribution #4 (Analysis): Shallow, incomplete, and lacking scientific insight

The paper claims to provide “analyses,” but the analytical component remains superficial:

What the paper does:

a) A simple Kendall correlation (Fig. 3)
b) A small IT-2AFC ablation (Table 4)
c) OOD performance comparison
d) N-way alternative test on a single dataset

What is missing:

a) No error analysis or qualitative failure study
b) No characterization of why encoder models generalize better than LMMs
c) No dataset-level statistical analysis

These omissions prevent the analysis from providing meaningful scientific insights into perceptual metrics.
Thus, the fourth contribution is overstated.

**Additional Comments:**

Nil

**Audience:**

Yes

**Audience Explanation:**

1) Practical Utility:
UniSim-Bench offers a single, unified evaluation framework for many perceptual similarity tasks, which is convenient for researchers and practitioners.

2) Engineering Convenience:
A ready-to-use multi-task perceptual metric (UniSim) built on CLIP/LLaVA is useful as a baseline, even without methodological novelty.

3) Industry Relevance:
Companies working on image generation, editing, and multimodal alignment may adopt this unified benchmark for internal evaluation.

4) Ecosystem Value:
The benchmark may become a reference point for future work on unified perceptual metrics, prompting discussion and follow-up studies.

**Broader Impact Concerns:**

Nil

**Claims And Evidence:**

No

**Claims Explanation:**

While the paper is well-written and the experimental results are extensive, the core contributions are either not novel, not well-supported, or not fairly evaluated:

1) The benchmark lacks principled design and analysis.

2) The “limitations of specialized models” claim is expected and experimentally unfair.

3) The UniSim model adds no methodological novelty.

4) The analyses are shallow and do not rise to the level expected for a top-tier venue.

**Requested Changes:**

1) Provide deeper dataset and task-level statistical analyses to justify the benchmark.

2) Conduct symmetry-fair comparisons, e.g., fine-tune baselines on the same subset of UniSim-Bench.

3) Add genuine analysis of multi-task learning, not just surface-level accuracy plots.

4) Include qualitative and representational analysis to understand what UniSim learns.

5) Either acknowledge the limited novelty of UniSim or introduce genuinely new methodological ideas.

---

> ### Author Response · Authors · 2026-01-10
> **Response to Reviewer zVa2**
>
> Dear Reviewer zVa2,
> Thank you for taking the time to review our paper. We appreciate your recognition of the value of standardizing perceptual tasks into a unified evaluation protocol, and we are pleased that you found the evaluations to be thorough.
>
> -------------------------------
>
> - Contribution #1 (UniSim-Bench): Insufficient originality and lack of deeper analysis
>
> - Provide deeper dataset and task-level statistical analyses to justify the benchmark.
>
> **Answer:**
> We would like to clarify that the primary goal of **UniSim-Bench** is not merely "mechanical aggregation," but to provide the first holistic framework that reveals significant **generalization gaps** in current specialized perceptual metrics that were previously obscured by task-specific research.
>
> 1. Principled Task Selection and Coherence
>
>  The seven tasks were selected to cover the fundamental dimensions of human perception in multi-modal contexts.
>
> **Core 2AFC Tasks (Img-2AFC, IT-2AFC, Text-2AFC, IQA):** these tasks share deep conceptual connections. For instance, `IT-2AFC` and `Text-2AFC` are inverse tasks that both measure cross-modal alignment. Furthermore, evaluating synthetic images—a primary motivation for these metrics—often requires simultaneously assessing both image quality (`IQA`) and alignment (`IT-2AFC`). Moreover, to compare the semantic similarity between the synthetic images and the references the `Img-2AFC` becomes important.
>
> **OOD Generalization Tasks (PAA, OOO, IR):** These were specifically chosen to test the boundaries of a metric’s generalization to different structures (IR) and semantics (PAA, OOO). We test structural generalization using Image Retrieval (IR), which forces the model to move from simple binary choices to ranking large pools of images. Moreover, we test semantic generalization using PAA and OOO, which challenge the model to understand specific visual attributes and logical groupings that differ from the core training data.
>
> 2. Analysis of Dataset Heterogeneity and Shifts
>
> Rather than focusing on static properties, we analyzed the **functional heterogeneity** of these datasets through our evaluation. Our results in **Table 2** and **Table 4** reveal that:
>
> **Intra-task Generalization Gaps:** Models trained on one dataset (e.g., HPDv2) often fail on another within the *same* task (e.g., HQ-EDIT), exposing significant distribution shifts in what humans value in "alignment" versus "quality".
>
>
> **Data Scaling:** We demonstrate that increasing the variety of datasets (e.g., adding MAGICBRUSH to HQ-EDIT) significantly improves intra-task robustness, quantifying the impact of dataset bias in existing specialized models.
>
>
> 3. Insights from Cross-Task Correlation (Fig. 3)
>
> The correlation heatmap in Fig. 3 provides a critical diagnostic tool:
>
> **Generalist vs. Specialized Perception:** Fig. 3a shows that general foundation models have a naturally correlated perceptual space, while Fig. 3b proves that **specialization (fine-tuning) actively degrades this coherence**, leading to negative correlations even between highly similar tasks like IT-2AFC and Text-2AFC.
>
> * This insight directly challenges the current trend of training "narrow" metrics and provides the first empirical evidence that a unified training objective (UniSim) is necessary to preserve general perceptual capabilities.
>
> Finally, we note that UniSim-Bench is designed as an **open-ended benchmark**. Its value lies in exposing the **"specialization-generalization trade-off"** in perceptual similarity metrics, providing a roadmap for developing metrics that mimic the broad, robust nature of human visual judgment.

---

> > ### Author Response · Authors · 2026-01-10
> >
> > - Contribution #2 (Limitations of specialized models): Conceptually trivial and empirically unsupported
> > - Conduct symmetry-fair comparisons, e.g., fine-tune baselines on the same subset of UniSim-Bench.
> >
> > **Answer:**
> >
> > **1. On the Novelty of "Lack of Generalization"**
> > The reviewer states that the lack of generalization in specialized metrics is "well-known." While we agree that this limitation is intuitively understood within the community, we argue that it has primarily existed as an assumption rather than a scientifically quantified fact. Prior to this work, **there has been no systematic, large-scale empirical study** measuring exactly *how much* performance degrades when specialized models are applied to closely related tasks. For example, while one might expect an Image-to-Text (IT-2AFC) model to struggle with Image Retrieval, it is scientifically significant to demonstrate that it also fails on the highly correlated Text-to-Image (Text-2AFC) task—performing worse than a zero-shot CLIP baseline (as seen in Figure 3 and Table 2).
> >
> > Furthermore, our study yields new, counter-intuitive insights regarding model architecture that go beyond simple data mismatches. By training both LMM-based and encoder-based models on the *same* data (UniSim), we conducted a strictly controlled comparison of these paradigms. We provide the critical empirical discovery that **encoder-based models generalize better for perceptual similarity than generative LMMs**, which tend to overfit to the structure of the training data. This finding challenges the assumption that larger generative models are inherently superior for all tasks and is crucial for guiding future architectural choices in the field.
> >
> >
> > **2. Addressing the "Unfair Comparison" and Distributional Shift**
> > The reviewer raises a concern that the comparison is not symmetry-fair because UniSim is trained on multiple tasks while baselines are specialized. We wish to clarify that the goal of our study is not to show that UniSim beats specialized models but to contrast the robustness of a **Unified Specialist** against **Narrow Specialists**. Regarding the fairness of the evaluation data:
> >
> > **Intra-Task Generalization (Unseen Datasets):** UniSim is evaluated on datasets it *never* saw during training (e.g., BAPPS, ImageReward, AGIQA-3K, CD-COCO, KONIQ-10K, as detailed in Section 3.3 and Table 2 ). In these cases, both UniSim and the specialized baselines are operating out-of-distribution regarding the specific dataset statistics. The results show that UniSim generalizes significantly better to these unseen domains than the specialized baselines do.
> >
> >
> > **OOD Generalization (Unseen Tasks):** We specifically included "OOD Generalization Tasks" (PAA, Odd-One-Out, IR) where **neither** UniSim nor the baselines were trained (Section 3.2 ). In Table 3, UniSim demonstrates superior or competitive performance compared to specialized baselines, proving that its multi-task training confers a general robustness that single-task training destroys.
> >
> >
> > **3. Scientific Value of the Baselines**
> > The reviewer suggests the poor performance of baselines is "expected" due to objective misalignment. We respectfully contend that demonstrating the *magnitude* of this misalignment is vital. It highlights a critical research gap: current "State-of-the-Art" metrics are often brittle and overfitted to specific datasets (e.g., DreamSim on NIGHTS). By benchmarking them on UniSim-Bench, we reveal that their dominance is highly conditional. This is not merely a "training-data mismatch" but a demonstration of the *fragility* of current metric design paradigms compared to a unified approach.

---

> > > ### Author Response · Authors · 2026-01-10
> > >
> > > - Contribution #3 (UniSim model): No methodological novelty UniSim is not a new model. It is:
> > >
> > > **Answer:**
> > > We note that the contribution of this work is **empirical and paradigmatic**, rather than architectural, trying to provide accurate, evidence-based insights into the performance of current SOTA perceptual metrics.
> > >
> > > **Simplicity as a Control Variable:** We intentionally employed standard architectures and loss functions to isolate the impact of our primary hypothesis: that **unified multi-task training** is superior to the prevailing paradigm of task-specific specialization. By keeping the architecture and loss function standard, we demonstrate that the improvements in performance and generalization stem directly from the *unified framework* rather than complex engineering or novel objectives.
> > >
> > >
> > > **Addressing the "Specialization Trap":** Prior work has treated perceptual tasks (e.g., Image-to-Text alignment vs. Image Quality Assessment) as disjoint problems requiring specialized models. Our work provides the first systematic evidence that this specialization leads to "poor inter-task generalization," where models fine-tuned on one task degrade on strongly correlated tasks.
> > >
> > >
> > > **Empirical Significance:** The significance lies in our analysis. Our standard fine-tuning recipe allows UniSim to outperform specialized state-of-the-art models (like DreamSim and HPSv2) on their own benchmarks while maintaining superior generalization. This is a valuable insight for the community, proving that standard VLMs are sufficient for high-performance perceptual metrics if the training data is unified correctly.
> > > Moreover, by training both LMM-based and encoder-based models on the same data, we conducted a controlled comparison of these paradigms. We provide the critical empirical discovery that encoder-based models generalize better for perceptual similarity than generative LMMs, which tend to overfit to the structure of the training data. This finding is crucial for guiding future architectural choices in the field.

---

> ### Author Response · Authors · 2026-01-10
>
> - Contribution #4 (Analysis): Shallow, incomplete, and lacking scientific insight
>
> **Answer:**
> We believe our manuscript provides specific scientific insights regarding these mechanisms, which we highlight below:
>
> **1. Characterization of Encoder vs. LMM Generalization**
>  We explicitly analyze why encoders generalize better in **Section 4.3** and **Section 4.4**.
> More specifically, we identify that LMMs (like LLaVA-NeXT) tend to overfit to the *structure* of the training data (i.e., the 2AFC "A vs. B" format). When tested on Out-Of-Distribution (OOD) tasks like Odd-One-Out (OOO), LMM performance drops significantly because the task structure changes.
> In contrast, we analyze that CLIP-based encoders are "unaffected as each input item is encoded independently regardless of the task". This is a key mechanistic insight: encoders succeed at generalization because they learn a robust embedding space, whereas LMMs learn a task-specific text-generation pattern.
>
> **Scaling to N-Alternatives:** We further supported this in Fig. 4, demonstrating minor as we increase the number of alternatives (N>2), LMM performance degrades while Encoders remain robust.
>
>
> **2. Insights on Negative Transfer**
> Our correlation analysis (Fig. 3) reveals a critical flaw in current metric learning: **negative transfer**. We show that specialized models exhibit weak or negative correlations across related tasks. For instance, metrics specialized in IT-2AFC fail to transfer to Text-2AFC despite the semantic similarity of the tasks. This analysis provides the "scientific insight" that fine-tuning on narrow perceptual distributions actively harms the model's zero-shot capabilities—a finding that justifies the need for the UniSim framework.
>
> **3. Scale and In-Context Learning Analysis**
> We went beyond standard benchmarking to investigate two critical dimensions of modern foundation models, yielding counter-intuitive scientific findings:
>
> Limits of Parameter Scaling (Section 4.4): We systematically evaluated the InternVL2.5 family across three scales (8B, 26B, and 38B). Contrary to the prevailing "scaling law" assumption in the field, we identify a performance plateau on Core 2AFC tasks and a regression on OOD tasks (e.g., PAA) at the 38B scale. This provides the specific insight that simply increasing model size is insufficient to capture fine-grained perceptual similarity.
>
> Ineffectiveness of In-Context Learning (Appendix C & Figure 5): We conducted a specific ablation on In-Context Learning (ICL) for LMMs (ranging from 1 to 3 shots). We report the negative result that ICL consistently reduces accuracy across all evaluated models. This analysis is crucial for the community, as it highlights a fundamental limitation in current ICL strategies when applied to rigorous perceptual comparison tasks, distinguishing them from standard reasoning tasks where ICL typically helps.
>
> These analyses isolate specific failure modes of current architectures, the fragility of LMMs to task structure, the negative transfer of specialized fine-tuning, and the inefficiency of scaling and ICL for perception. We believe these constitute concrete scientific contributions to understanding multi-modal metrics.

---

> > ### Author Response · Authors · 2026-01-10
> >
> > - Add genuine analysis of multi-task learning, not just surface-level accuracy plots.
> >
> > **Answer:**
> > We  respectfully direct the reviewer to several existing sections of our paper where we conduct deep analyses to investigate the multi-task learning mechanics:
> >
> > **1. Analysis of Task Correlation and Feature Degradation (Section 3.2 & Figure 3)**
> > We explicitly analyzed the structural relationship between tasks by computing Kendall’s correlation of model performance across task pairs.
> >
> > * **Specialized vs. Generalist Dynamics:** As illustrated in Figure 3, we show that specialized models often exhibit weak or negative correlations with other tasks, indicating that fine-tuning on narrow tasks degrades the general feature space (catastrophic forgetting).
> >
> >
> > * **Multi-task Learning (MTL) Benefit:** Conversely, our analysis shows that general-purpose models maintain positive correlations across core tasks, providing analytical evidence that our MTL formulation preserves the semantic alignment necessary for broad generalization.
> >
> >
> > **2. Analysis of Positive Transfer via Data Ablation (Section 4.4 & Table 4)**
> > In Section 4.4, we went beyond reporting final accuracy by conducting an ablation study on the IT-2AFC task training data.
> >
> > * **Mechanism of Improvement:** We analyzed how varying the number of training datasets affects not only intra-task generalization but also inter-task performance.
> >
> >
> > * **Cross-Task Benefits:** Table 4 demonstrates that jointly training on multiple datasets yields positive transfer to distinct but related tasks (e.g., Text-2AFC), proving that the performance gains stem from mutual benefits in the multi-task formulation rather than just data scaling.
> >
> >
> > **3. Analysis of Generalization Boundaries (Section 4.3)**
> > We provide a granular analysis of *where* MTL succeeds and fails regarding Out-Of-Distribution (OOD) tasks.
> >
> > * **Near-OOD vs. Far-OOD:** We distinguish between "Near-OOD" tasks (like PAA), where multi-task training is beneficial due to structural similarity, and "Far-OOD" tasks (like Odd-One-Out), where the benefit diminishes.
> >
> >
> > * **Architecture Analysis:** We further analyze why LMM-based models struggle more than CLIP-based models on specific OOD tasks (e.g., OOO), attributing it to the structural differences between the 2AFC training data and the target task, which affects generative models more severely.
> >
> >
> > We believe these sections (3.2, 4.3, and 4.4) collectively provide a genuine analysis of the multi-task learning dynamics, covering task inter-dependency, positive transfer, and generalization limits.
> >
> > -------------------------------
> >
> > - Include qualitative and representational analysis to understand what UniSim learns.
> >
> > **Answer:**
> > In the paper, we include specific analyses to investigate **the underlying representations learned by UniSim** by examining the compositionality and robustness of its features. We specifically direct attention to the following two analyses as evidence of the model's learned representations:
> >
> > * **Cross-Task Feature Learning (Section 4.4 & Table 4):** We conducted an ablation study to understand the compositionality of the learned features by varying the training data for the IT-2AFC task. As detailed in Table 4, training on multiple datasets not only improves intra-task generalization but also enhances performance on a distinct task, Text-2AFC. This indicates that UniSim learns high-level, transferable perceptual features that are mutually beneficial across modalities, rather than simply memorizing dataset-specific patterns.
> >
> >
> > * **UniSim performance in more difficult inference settings (Section 4.4 & Figure 4):** We probed the quality and density of the learned embedding space by analyzing the effect of increasing the number of alternative images from 2 to  (up to 8) at test time. As shown in Figure 4, UniSim significantly outperforms the CLIP baseline and maintains high accuracy even as the number of distractors increases. This confirms that the model learns a robust, discriminative metric space capable of distinguishing fine-grained differences, rather than a brittle binary decision boundary.
> >
> >
> > We believe these behavioral analyses provide a concrete, objective understanding of the UniSim **models' strengths and weaknesses**.
> >
> > -------------------------------
> >
> > - Either acknowledge the limited novelty of UniSim or introduce genuinely new methodological ideas.
> >
> > **Answer:** As noted in the general response, we have revised the limitations section to provide a more comprehensive discussion of the study’s limitations.

---

### Review · Reviewer_yVKW · 2025-12-14

**Summary Of Contributions:**

### Summary
The authors propose a new benchmark to combine perceptual similaritiy tasks under one unified approach. The findings suggest that specialised models don't generalise well to other tasks. The authors then propose UniSim, a model designed to be more generalisable across tasks

### Strengths

1. It tackles a real problem and addresses a real gap. Perceptual similarity vary between benchmarks and the terminology and definitions aren’t unified.
2. The experiments are quite comprehensive with 12 models and multiple datasets
3. Overall, the paper is well written and easy to follow and well organised.

### Weaknesses
1. My biggest issue is that there seems to be conflation between simply training on more data vs using better methods. UniSim trains on a superset of data compared to the specialised models. To understand where the performance gain comes from, some sort of experiment is needed, for example an ablation study such as CLIP fine tuned on all data for a single task.
2. The held out datasets used arent truly OOD in a generalisation sense. They seem to be from the same task in the same format (2AFC) so this is more of a test of dataset shift rather than distribution shift / generalisation.
3. It seems that on different tasks, UniSim can hurt performance. Is this correctly interpreted? It seems training on 2AFC tasks damages performance on non-2AFC tasks. Clip at 59% and Unisim at 48% in Table 3. The authors acknowledge this but don’t discuss the implications.
4. The evaluation protocol isn’t very clear, in the appendix (page 21) I see that two approaches are tested and the best one is picked. This has a few issues. First, this means there is test set leakage since the model selection is the same data as presented as “test set”. A proper way would be to then apply the best performing approach to a final held out set. This also affects reproducibility as the methods in the results haven’t used the same approach. Presenting both equally would be a fairer comparison.
5. Data from HQ-Edit is in both IT-2AFC AND Text-2AFC, which means these tasks aren’t truly independent and claims of generalisation across these specific tasks are confounded
6. There also seems to be inconsistency and confusion between the terms “metric” and “model”, which are used interchangeably. UniSim is called both a set of metrics and other times a set of models. A model can be used to define a metric, but this is not very clear in the paper. A metric should be well-defined. Also, and perhaps	related, Table 2 seems to compare very different things as equivalent: CLIP (zero shot embedding similarity, a metric) and LLaVA prompted text generation (not a metric) and what’s missing is how each system computes 2AFC decision.
7. The limitations section is pretty superficial and doesn’t sufficiently cover the specific limitations of this work. There are many specific limitations here, just a few would be: Eval protocol inconsistency, no statistical significance testing, held out sets aren’t fully held out, tasks share datasets in some instances.

**Audience:**

Yes

**Audience Explanation:**

A new benchmark and further unification of multi-modal tasks is of interest and there is a need for more research in the field.

**Broader Impact Concerns:**

No special ethical concerns noted.

**Claims And Evidence:**

No

**Claims Explanation:**

1. Many of the claims are supported, especially the general claims around the fact that specialised models don’t generalise well. However, some of the more detailed claims, especially around UniSim are not supported.
2. For example, DreamSim Ensemble achieves a higher score than UniSim on Img-2AFC, this is acknowledged briefly but downplayed and UniSim is mostly compared against weaker variants and models.
3. UniSim as a “unified metric” isn’t very supported. UniSim can hurt OOD performance (See weakness 3)
4. The claim that multi-task training improves performance isn’t clearly supported since it’s not isolated from the effects of training on more data

I just want to highlight here for the authors' revision here that SOTA scores are not a requirement for TMLR but the claims must be backed by the evidence.

**Requested Changes:**

Critical:
1. Ablation/isolation test to see if improved performance is coming from more data volume vs different tasks
2. Investigate why OOD performance degrades. Similarly, adjust the claims on a “unified metric” or add a clear disclaimer/limitation.
3. Use consistent evaluation protocols for all models, or very clearly specify what protocol was used for each model in the tables.
4. Improve the limitations section

Less critical:
1. Better define “model” vs “metric” in the context of this paper
2. Add statistical significance testing (error bars / confidence intervals)

---

> ### Author Response · Authors · 2026-01-10
> **Response to Reviewer yVKW**
>
> Dear Reviewer yVKW,
> Thank you for taking the time to review our paper. We appreciate that you recognized the importance of the problem and the current lack of a unified framework for perceptual metrics.
>
> -------------------------------
>
> - My biggest issue is that there seems to be conflation between simply training on more data vs using better methods. UniSim trains on a superset of data compared to the specialised models. To understand where the performance gain comes from, some sort of experiment is needed, for example an ablation study such as CLIP fine tuned on all data for a single task.
>
> - Ablation/isolation test to see if improved performance is coming from more data volume vs different tasks
>
>
> **Answer:** We clarify that UniSim **does not train on a superset of data compared to specialized models**. To prevent task dominance, we strictly balance our training distribution, capping each task at 400K samples.
>
> Consequently, for major tasks, UniSim actually **sees less data than the specialized baselines**:
> UniSim vs. HPSv2 (IT-2AFC): The specialized HPSv2 model is trained on the full HPDv2 dataset (~798K pairs). In contrast, UniSim is trained on a downsampled mixture of datasets capped at 400K for this task. Despite seeing fewer task-specific samples, UniSim achieves superior cross-task generalization.
>
> We believe the "single-task" experiments the reviewer requested are effectively represented by the Specialized Models (DreamSim, HPSv2, PAC-S) reported in Table 2. These are SOTA models trained specifically on their respective large-scale datasets. Our results demonstrate that while these models perform well on their specific training task, they suffer from negative transfer on related tasks (Figure 3).
>
> Finally, we explicitly investigated the **"data vs. method" question in Table 4**, where we ablated the training data for the IT-2AFC task. The results show that increasing dataset diversity (adding HQ-EDIT + HPDv2) yields significant gains in generalization to unseen datasets (ImageReward) compared to training on single datasets, confirming that the multi-task formulation—not just data volume—is the primary driver of performance.
>
> -------------------------------
>
> - The held out datasets used arent truly OOD in a generalisation sense. They seem to be from the same task in the same format (2AFC) so this is more of a test of dataset shift rather than distribution shift / generalisation.
>
> **Answer:** We would like to clarify that our benchmark, **UniSim-Bench**, explicitly evaluates both types of generalization, organized into two distinct tiers as detailed in **Section 3.3**:
>
> **1. Robustness to Dataset Shift (Intra-Task Generalization):**
> The reviewer is correct that for the held-out datasets within the **Core 2AFC Tasks** (e.g., training on HPDv2 and testing on ImageReward), the format remains 2AFC. We acknowledge that this specifically measures robustness to **dataset shift**—verifying that the model does not overfit to specific annotation styles or image domains within a known task structure.
>
> **2. True OOD Generalization (Inter-Task & Structural Shift):**
> To address the "generalization sense" the reviewer mentions, we included a separate category called **OOD Generalization Tasks** (Section 4.3, Table 3), which were **completely excluded from training**. Critically, these tasks differ not just in data distribution, but in task definition and input structure:
>
>
> **Odd-One-Out (OOO):** This task uses a triplet format where the goal is to identify the outlier. This is structurally distinct from the 2AFC preference pairs used during training.
>
>
> **Image Retrieval (IR):** This task involves ranking a query against a large pool of images, which fundamentally differs from the binary classification format of the training phase.
>
>
> **Perceptual Attributes Assessment (PAA):** While this uses a pairwise format, it tests specific attribute recognition (e.g., "brightness", "colorfulness") that the model was not explicitly supervised on, representing a semantic shift.

---

> > ### Author Response · Authors · 2026-01-10
> >
> > - It seems that on different tasks, UniSim can hurt performance. Is this correctly interpreted? It seems training on 2AFC tasks damages performance on non-2AFC tasks. Clip at 59% and Unisim at 48% in Table 3. The authors acknowledge this but don’t discuss the implications.
> >
> > - Investigate why OOD performance degrades. Similarly, adjust the claims on a “unified metric” or add a clear disclaimer/limitation.
> >
> > **Answer:**
> > We confirm the reviewer's observation that UniSim's performance drops on the average of OOD Generalization Tasks (48.8% vs. 59.4% for ViT-L/14 in Table 3).
> >
> > However, we discuss the implications of this in **Section 4.3**, where we distinguish between task structures:
> >
> > 1. **Far-OOD Tasks (OOO & IR):** We attribute the performance drop specifically to the **Odd-One-Out (OOO)** and **Image Retrieval (IR)** tasks. These tasks differ structurally from the pairwise (2AFC) training objective, causing the model to overfit to the binary comparison format.
> >
> >
> > 2. **Near-OOD Tasks (PAA):** Conversely, for **Perceptual Attributes Assessment (PAA)**—which we define as 'near-OOD' because it retains the pairwise comparison structure—UniSim actually **outperforms** the CLIP baseline (e.g., UniSim ViT-L/14 achieves 67.6% vs. CLIP's 65.8%).
> >
> >
> > Therefore, the implication is that while UniSim effectively learns perceptual features (improving performance on unseen tasks with similar structures like PAA), the fine-tuning process limits the model's ability to generalize to completely different task interfaces (like ranking or triplet selection).
> >
> > -------------------------------
> >
> > - The evaluation protocol isn’t very clear, in the appendix (page 21) I see that two approaches are tested and the best one is picked. This has a few issues. First, this means there is test set leakage since the model selection is the same data as presented as “test set”. A proper way would be to then apply the best performing approach to a final held out set. This also affects reproducibility as the methods in the results haven’t used the same approach. Presenting both equally would be a fairer comparison.
> >
> >
> > - Use consistent evaluation protocols for all models, or very clearly specify what protocol was used for each model in the tables.
> >
> > **Answer:** The reviewer's concern regarding test set leakage is misplaced because the evaluation of baselines involves choosing the optimal zero-shot inference strategy (which requires no parameter training), while the proposed UniSim models follow a fixed, pre-defined protocol. For more clarification, we provide the following information:
> >
> > **Zero-Shot Baseline Representation:** For general-purpose baselines (e.g., CLIP), we evaluated two common zero-shot approaches—**naive alignment** and **CLIP-IQA**—to report the most competitive baseline for each model. This is intended to show the "upper bound" of these models' zero-shot capabilities rather than to optimize parameters on the test set.
> >
> >
> > **UniSim Consistency:** Our proposed **UniSim models** do not rely on this "best-of-two" selection; instead, they follow a fixed evaluation protocol defined by the multi-task training objectives and the specific instructions detailed in **Table 7**.
> >
> >
> > **Test Set Leakage:** Since no model parameters were updated or fine-tuned based on these test results, and because the UniSim models utilize a singular, pre-defined approach (Table 7), the risk of traditional test set leakage is minimal. However, we agree that presenting both methods equally is a more rigorous standard.
> >
> > -------------------------------
> >
> > - Data from HQ-Edit is in both IT-2AFC AND Text-2AFC, which means these tasks aren’t truly independent and claims of generalisation across these specific tasks are confounded
> >
> >
> > **Answer:** We contend that these tasks are conceptually and empirically distinct, and **our claims of generalization remain robust** for the following reasons:
> >
> > 1- While both tasks utilize the same source content, the direction of alignment is inverted. **IT-2AFC** evaluates the model’s ability to select the best **image** for a given prompt, whereas **Text-2AFC** evaluates the selection of the best **caption** for a given image.
> >
> >
> > 2- If the tasks were confounded, models specialized in one should naturally excel in the other. Empirically, this is not the case: the **HPSv2** model (specialized for IT-2AFC) performs worse than its general-purpose baseline on the **Text-2AFC** task (68.9% vs. 72.0%), despite the shared data domain.
> >
> >
> > 3- Our claims of generalization are further supported by performance on datasets entirely **excluded from training**. For instance, UniSim achieves superior results on **CD-Coco** (Text-2AFC) and **AGIQA-3K** (IT-2AFC), which were never seen by the model during fine-tuning.

---

> > > ### Author Response · Authors · 2026-01-10
> > >
> > > - There also seems to be inconsistency and confusion between the terms “metric” and “model”, which are used interchangeably. UniSim is called both a set of metrics and other times a set of models. A model can be used to define a metric, but this is not very clear in the paper. A metric should be well-defined. Also, and perhaps related, Table 2 seems to compare very different things as equivalent: CLIP (zero shot embedding similarity, a metric) and LLaVA prompted text generation (not a metric) and what’s missing is how each system computes 2AFC decision.
> > >
> > > - Better define “model” vs “metric” in the context of this paper
> > >
> > >
> > > **Answer:**
> > >
> > > In this work, we refer to UniSim as a family of models (specifically encoder-based and generative Vision-Language Models) that are used as perceptual metrics. **A "metric" in our context is the specific function that translates model outputs into a similarity score or a preference decision.** We will standardize the text to reflect that UniSim models serve as a perceptual metric.
> > >
> > > The exact mechanisms for how each category of model computes a 2AFC decision are defined in the following locations in the paper:
> > >
> > > **Encoder-based Models (e.g., CLIP-based UniSim):** The metric is defined in **Section 2.2 (Lines 127–130)** and **Section 3.3 (Lines 271–274)**. These models quantify similarity via the **cosine similarity** of embedding vectors. For a reference and two alternatives, the 2AFC decision is determined by the similarity function , selecting the alternative that yields the highest value.
> > >
> > >
> > > **Generative Models (e.g., LLaVA-based UniSim):** The decision mechanism is defined in **Section 2.2 (Lines 134–137)** and **Section 3.3 (Lines 286–289)**. These models utilize **natural language prompting** to solve 2AFC tasks. The model is presented with images and a prompt (e.g., "Which image is better described by...?") and produces a single-word prediction between the two alternatives.
> > >
> > >
> > > **Comparability in Table 2**
> > > While the internal mechanisms (embedding similarity vs. text generation) differ, Table 2 compares them as equivalent "perceptual observers". Both systems are tasked with a binary classification problem: selecting the alternative that human annotators preferred. We have standardized all datasets into this **2AFC format** specifically to allow this direct comparison of classification accuracy.
> > >
> > > -------------------------------
> > >
> > > - The limitations section is pretty superficial and doesn’t sufficiently cover the specific limitations of this work.
> > >
> > > - Improve the limitations section
> > >
> > > **Answer:** As noted in the general response, we revised the limitations section to provide a more comprehensive discussion of the study’s limitations.
> > >
> > > -------------------------------
> > >
> > > - Add statistical significance testing (error bars / confidence intervals)
> > >
> > > **Answer:**
> > > We agree that assessing statistical significance is crucial for validating the robustness of our results. In the revised manuscript, we have calculated 95% confidence intervals for the Core 2AFC Tasks and included them in Table 6 (Appendix C). These intervals were computed based on the exact sample size of each test set. This analysis confirms that UniSim achieves statistically significant improvements over the baselines on the majority of tasks. Furthermore, the addition of explicit error bars provides the reader with a clearer interpretation of the performance margins.

---

### Review · Reviewer_d885 · 2025-12-20

**Summary Of Contributions:**

This paper addresses the fragmentation of perceptual similarity evaluation across uni- and multi-modal tasks by proposing UniSim-Bench, a unified benchmark that consolidates seven perceptual similarity tasks spanning 25 datasets. By reformulating diverse tasks under a common 2AFC-based evaluation protocol and distinguishing between core tasks and out-of-distribution (OOD) generalization tasks, the benchmark enables systematic comparison of perceptual metrics across modalities and task settings.

Using UniSim-Bench, the authors conduct a large-scale empirical study of general-purpose vision–language models, task-specific perceptual metrics, and newly trained multi-task models. The evaluation reveals that many specialized metrics, while strong on their target datasets, exhibit limited intra-task and inter-task generalization, and in some cases underperform general-purpose models on unseen datasets or related tasks. The study further shows that encoder-based vision–language models tend to generalize more reliably as perceptual metrics than generative multimodal models, particularly under task distribution shifts.

As an initial step toward unified perceptual metrics, the paper introduces UniSim, a family of multi-task perceptual models trained on a subset of the benchmark’s core tasks. While the UniSim models rely on relatively standard training objectives, they achieve improved average performance and stronger generalization across tasks compared to both general-purpose and specialized baselines, illustrating the potential benefits of multi-task training in this domain.

Key Strengths:

1. The paper introduces a comprehensive and well-structured benchmark for evaluating perceptual similarity metrics across multiple modalities and tasks, addressing a clear gap in existing evaluation practices.

2. The experimental evaluation is extensive and carefully controlled, covering a broad range of model families and explicitly analyzing different forms of generalization.

3. The paper provides useful empirical insights into the limitations of task-specific perceptual metrics and highlights systematic differences between encoder-based and generative vision–language models.

4. Claims are stated in a measured and cautious manner, aligning well with the evidence presented.

Key Weaknesses:

1. The modeling contribution (UniSim) is incremental, and largely serves as a strong empirical baseline rather than introducing novel architectural or training innovations.

2. While the paper convincingly demonstrates generalization failures and task interference, the analysis remains primarily empirical, with limited insight into the underlying representational or mechanistic causes.

3. The notion of task relatedness and out-of-distribution generalization is discussed post hoc, and could benefit from a more principled or formal characterization.

**Additional Comments:**

Overall, the paper is clearly written and well organized, with a logical progression from problem motivation to benchmark design and empirical analysis. The figures and tables are generally informative, and the separation between core tasks and out-of-distribution generalization tasks is helpful for structuring the evaluation.

One particularly positive aspect is the authors’ careful and restrained interpretation of results. Rather than overclaiming the success of a unified perceptual metric, the paper consistently emphasizes remaining challenges and limitations, which contributes to the credibility of the conclusions.

From a presentation perspective, the paper could further benefit from short, high-level summaries at the end of dense experimental sections to guide readers toward the main takeaways. This would improve readability without requiring additional experiments.

Finally, UniSim-Bench appears to be a valuable resource for the community, and its open-ended design suggests potential for future extensions. Making the benchmark easily extensible and clearly documenting dataset splits and evaluation protocols will be important for maximizing its long-term impact.

**Audience:**

Yes

**Audience Explanation:**

The findings of this paper are likely to be of interest to a substantial portion of TMLR’s audience, particularly researchers working on evaluation, perceptual metrics, representation learning, and multimodal vision–language models. The paper addresses a long-standing and practically relevant issue: the lack of unified evaluation frameworks for perceptual similarity across different modalities and tasks.

By systematically demonstrating the limitations of task-specific perceptual metrics in terms of generalization, and by highlighting consistent differences between encoder-based and generative vision–language models, the paper provides insights that are broadly relevant beyond any single application or dataset. The introduction of a unified benchmark further enhances the paper’s relevance, as it offers a reusable resource for future research on perceptual alignment and model evaluation.

Even though the modeling contribution is modest, the empirical findings and benchmark design provide useful guidance for researchers developing or applying perceptual metrics, making the paper aligned with the interests and goals of the TMLR community.

**Broader Impact Concerns:**

The paper focuses on benchmarking and evaluating perceptual similarity metrics across uni- and multi-modal tasks, and does not introduce new data collection procedures or deploy models in high-stakes decision-making contexts. As such, the work does not raise immediate or severe ethical concerns.

Potential broader impact considerations primarily relate to the downstream use of perceptual metrics in applications such as generative model evaluation, content filtering, or model alignment. In such contexts, biases or inconsistencies present in the underlying datasets could be amplified if the metrics are used uncritically. However, the paper explicitly highlights the limitations of current perceptual metrics and emphasizes generalization failures, which serves to caution against over-reliance on any single automated metric.

The benchmark itself aggregates existing datasets and does not introduce new human annotations, reducing risks related to privacy or consent. While a more explicit discussion of dataset biases and annotation variability could further strengthen the paper, the absence of a dedicated Broader Impact Statement does not appear to be a barrier to acceptance.

**Claims And Evidence:**

Yes

**Claims Explanation:**

The main claims of the paper are supported by extensive and carefully designed empirical evidence. The authors evaluate a broad set of general-purpose vision–language models, task-specific perceptual metrics, and multi-task models across a unified benchmark that spans multiple modalities, tasks, and datasets. The experimental setup explicitly distinguishes between standard train–test generalization, intra-task generalization to unseen datasets, and inter-task or out-of-distribution generalization, which aligns well with the claims made in the paper.

Quantitative results are reported consistently across all tasks, and the conclusions regarding the limited generalization of specialized perceptual metrics, as well as the relative robustness of encoder-based models compared to generative models, are directly supported by the presented results. Additional analyses, such as task correlation measurements and ablation studies on training data composition, further strengthen the empirical basis of the claims.

While some explanatory interpretations—particularly regarding the underlying causes of task interference and model generalization behavior—remain largely empirical and could benefit from deeper mechanistic analysis, the paper does not overstate these interpretations. Overall, the evidence provided is accurate, convincing, and sufficiently clear to support the claims as stated.

**Requested Changes:**

Critical Changes:

1. Clarify the conceptual justification for unifying the selected tasks under a single perceptual similarity framework.
While the paper successfully unifies multiple perceptual tasks under a common 2AFC evaluation protocol, the conceptual argument that these tasks share a common underlying notion of perceptual similarity remains largely implicit. The current justification relies primarily on empirical correlations and transfer results. A clearer discussion of why these tasks are expected to share representational structure—beyond evaluation convenience—would strengthen the foundation of the benchmark and reduce potential ambiguity regarding the scope of the claims.

2. Strengthen the analysis explaining task interference and generalization failure.
The paper convincingly demonstrates that specialized metrics often fail to generalize across datasets and tasks. However, the analysis remains largely descriptive. Providing deeper insight into the causes of task interference—such as modality asymmetry, supervision mismatch, or representation drift—would significantly improve the interpretability of the findings and help readers understand the observed performance trends beyond empirical outcomes.

Non-Critical Changes (Would Strengthen the Work):

1. Provide a more principled characterization of task relatedness and OOD difficulty.
The distinction between “near-OOD” and “far-OOD” tasks is introduced post hoc based on empirical performance. Introducing a more explicit or principled notion of task distance—either conceptually or empirically—would improve clarity and make the discussion of generalization more rigorous.

2. Expand discussion of annotation quality and human agreement across datasets.
Given the benchmark’s focus on human perceptual similarity, additional discussion of annotation consistency, inter-annotator agreement, or known dataset biases would help contextualize the results and clarify whether observed generalization failures stem from model limitations or dataset characteristics.

3. Clarify the positioning of UniSim as a baseline rather than a novel modeling contribution.
While UniSim performs well empirically, its modeling components largely build on established techniques. Explicitly emphasizing UniSim’s role as a strong multi-task baseline enabled by the benchmark—rather than a novel architecture—would help align reader expectations and avoid potential misinterpretation of the contribution.

4. Minor clarity improvements in presentation.
Some sections with dense tables could benefit from brief summary statements highlighting the most salient trends, helping readers more easily extract the key empirical insights without exhaustive cross-referencing.

---

> ### Author Response · Authors · 2026-01-10
> **Response to Reviewer d885**
>
> Dear Reviewer d885,
> Thank you for taking the time to review our paper. We appreciate that you recognized UniSim-Bench as a comprehensive benchmark addressing a clear gap and found our experimental analysis valuable.
>
> -------------------------------
>
> - The modeling contribution (UniSim) is incremental, and largely serves as a strong empirical baseline rather than introducing novel architectural or training innovations.
>
> - Clarify the positioning of UniSim as a baseline rather than a novel modeling contribution. While UniSim performs well empirically, its modeling components largely build on established techniques. Explicitly emphasizing UniSim’s role as a strong multi-task baseline enabled by the benchmark—rather than a novel architecture—would help align reader expectations and avoid potential misinterpretation of the contribution.
>
> **Answer:** We acknowledge that UniSim utilizes standard architectures; however, we emphasize that the primary contributions of our work are methodological and empirical rather than architectural:
>
> **Unification of Disjoint Tasks:** Prior work has largely treated perceptual tasks (e.g., IQA, image-text alignment) as disjoint problems requiring specialized models. We provide the first framework demonstrating that a single unified metric can competitively solve these diverse tasks, whereas specialized models often fail outside their narrow domains.
>
> **Encoder vs. LMM Analysis:** By training both LMM-based and encoder-based models on the same data, we conducted a controlled comparison of these paradigms. We provide the critical empirical discovery that encoder-based models generalize better for perceptual similarity than generative LMMs, which tend to overfit to the structure of the training data. This finding is crucial for guiding future architectural choices in the field.
>
> -------------------------------
>
> - While the paper convincingly demonstrates generalization failures and task interference, the analysis remains primarily empirical, with limited insight into the underlying representational or mechanistic causes.
>
> - Strengthen the analysis explaining task interference and generalization failure. The paper convincingly demonstrates that specialized metrics often fail to generalize across datasets and tasks. However, the analysis remains largely descriptive. Providing deeper insight into the causes of task interference—such as modality asymmetry, supervision mismatch, or representation drift—would significantly improve the interpretability of the findings and help readers understand the observed performance trends beyond empirical outcomes.
>
> **Answer:**
> While our primary contribution is an empirical benchmark, we do investigate specific architectural and representational causes for the observed failures in Sections 3.2, 4.2, and 4.3:
>
> 1. In Section 4.3, we provide evidence that Large Multi-modal Models (LMMs) fail on Out-of-Distribution tasks (like OOO) because they overfit to the specific **task structure** (e.g., 2AFC question formats). We contrast this with encoder-based models, which remain robust because they encode input items independently regardless of the task setup.
>
>
> 2. In Section 4.2, we identify why vision-only specialization (e.g., DreamSim) hurts performance on multi-modal tasks. We attribute this to a specific representational trade-off, where fine-tuning on vision-only data adversely impacts the learned image-text alignment.
>
> 3. Our correlation analysis in Section 3.2 (Fig. 3) reveals that while general-purpose models show positive correlations across tasks, specialized fine-tuning destroys these relationships. This indicates that the underlying cause of poor generalization is the deterioration of the model's zero-shot ability in the original feature space.

---

> > ### Author Response · Authors · 2026-01-10
> >
> > - The notion of task relatedness and out-of-distribution generalization is discussed post hoc, and could benefit from a more principled or formal characterization.
> >
> > - Provide a more principled characterization of task relatedness and OOD difficulty. The distinction between “near-OOD” and “far-OOD” tasks is introduced post hoc based on empirical performance. Introducing a more explicit or principled notion of task distance—either conceptually or empirically—would improve clarity and make the discussion of generalization more rigorous.
> >
> >
> > **Answer:** We acknowledge that our definition of task-relatedness is primarily empirical. However, we ground our characterization of task-relatedness and generalization in two concrete analyses:
> >
> > **Quantitative Correlation:** We utilized **Kendall’s correlation** (Fig. 3) to validate our task groupings. This analysis confirmed that for the foundation models, "Core" tasks are positively correlated with one another, whereas "OOD" tasks exhibit no correlation with the core group, empirically justifying their designation as out-of-distribution.
> >
> > **Structural Distinction:** We defined generalization based on task structure and training exclusion. Specifically, in **Section 4.3**, we distinguished between "near-OOD" tasks (PAA), which share the 2AFC structure of the training data, and "far-OOD" tasks (OOO, IR), which rely on fundamentally different structures like triplet ranking or retrieval.
> >
> > -------------------------------
> >
> > - Clarify the conceptual justification for unifying the selected tasks under a single perceptual similarity framework. While the paper successfully unifies multiple perceptual tasks under a common 2AFC evaluation protocol, the conceptual argument that these tasks share a common underlying notion of perceptual similarity remains largely implicit. The current justification relies primarily on empirical correlations and transfer results. A clearer discussion of why these tasks are expected to share representational structure—beyond evaluation convenience—would strengthen the foundation of the benchmark and reduce potential ambiguity regarding the scope of the claims.
> >
> >
> > **Answer**: Our unification of these tasks under a single framework is driven by the hypothesis that they represent distinct but interconnected facets of human perception, rather than disjoint problems.
> >
> > We offer the following conceptual justifications for this design:
> >
> > **Inherent Task Interconnectedness:** The selected tasks share deep conceptual connections. For instance, `IT-2AFC` and `Text-2AFC` are inverse tasks that both measure cross-modal alignment. Furthermore, evaluating synthetic images—a primary motivation for these metrics—often requires simultaneously assessing both image quality (`IQA`) and alignment (`IT-2AFC`).
> >
> >
> > **Shared Representational Structure:** Our empirical results confirm that these tasks rely on shared representations. For example, our ablation in Table 4 demonstrates that training on `IT-2AFC` improves performance on `Text-2AFC`, indicating that optimizing for one task enhances the underlying features needed for the other.
> >
> >
> > **The Necessity of Generalization:** A metric that captures true human perception must generalize across datasets and related perceptual similarity tasks. We argue that a single metric is necessary because existing specialized models fail to generalize even within the same task (e.g., specialized `IT-2AFC` models failing on `HQ-EDIT`). Unifying these tasks forces the model to learn robust, generalized features rather than overfitting to specific task artifacts.
> >
> > In summary, we unify these tasks because they rely on the same vision-language alignment features. A single metric with strong intra- and inter-task generalization is the most viable step toward broadly capturing the holistic "human notion of similarity".

---

> > > ### Author Response · Authors · 2026-01-10
> > >
> > > - Expand discussion of annotation quality and human agreement across datasets. Given the benchmark’s focus on human perceptual similarity, additional discussion of annotation consistency, inter-annotator agreement, or known dataset biases would help contextualize the results and clarify whether observed generalization failures stem from model limitations or dataset characteristics.
> > >
> > > **Answer:** We agree that annotation reliability is critical for interpreting generalization failures. As detailed in **Appendix B.1**, we implemented strict filtering protocols to maximize inter-annotator agreement and minimize ambiguity in our ground truth:
> > >
> > > **High-Confidence Filtering:** To reduce label noise, we focused on "confident" comparisons. For **ImageReward**, we restricted triplets to comparisons between the highest and lowest-ranked images. Similarly, for **CD-Coco**, we pruned samples with negative human votes , and for **AGIQA-3K**, we filtered out images with low perceptual quality scores prior to annotation.
> > >
> > >
> > > **Scale and Validation:** We selected datasets with robust statistical backing, such as **HPDv2** (798k human preference annotations) and **PIPAL** (1.13 million judgments), specifically to ensure "reliable subjective quality scores".
> > >
> > > Because our benchmarking data prioritizes these high-agreement scenarios rather than borderline cases, the observed lack of generalization is likely attributable to model limitations (e.g., overfitting to training distributions) rather than dataset inconsistencies.
> > >
> > > -------------------------------
> > >
> > > - Minor clarity improvements in presentation. Some sections with dense tables could benefit from brief summary statements highlighting the most salient trends, helping readers more easily extract the key empirical insights without exhaustive cross-referencing.
> > >
> > > **Answer:**
> > > As requested by the reviewer, we added brief "Observations" to the captions for Table 2 and Table 3 that summarize the salient trends already discussed in Sections 4.2 and 4.3.
> > >
> > > **Revised Caption for Table 2 (Evaluation on Core 2AFC Tasks):**
> > >
> > > > *Table 2: ... [Existing description]. **Observations:** (1) **Intra-task generalization:** Specialized models (e.g., HPSv2, ImageReward) often struggle on unseen datasets within their domain (e.g., HQ-EDIT), whereas UniSim generalizes successfully. (2) **Inter-task generalization:** Specialized models suffer performance degradation on tasks outside their training domain, while UniSim consistently achieves top-tier performance across diverse tasks.*
> > >
> > > **Revised Caption for Table 3 (Evaluation on OOD Generalization Tasks):**
> > >
> > > > *Table 3: ... [Existing description]. **Observations:** (1) **Generalization Gap:** Perceptual metrics generally underperform general-purpose baselines on out-of-distribution tasks (OOO, IR). (2) **Model Architecture:** Encoder-based models (CLIP) demonstrate stronger OOD generalization compared to LMMs, which tend to overfit to the training task structure.*

---

### Author Response · Authors · 2026-01-10
**General Response to Reviewers**

We sincerely thank the reviewers for their valuable and constructive feedback, which has helped us improve the clarity and quality of the manuscript. In response to the comments, we have made the following revisions:

1. We expanded the *Limitations* section to include two dedicated paragraphs discussing the limitations of the UniSim benchmarks and UniSim models. This expansion also incorporates the limitations highlighted during the rebuttal discussion.

2. For the performance reported on the Core-2AFC task, we now provide the 95% confidence interval values as subscripts in Table 6 to improve statistical transparency.

3. We added concise summaries of key observations to the captions of Table 2 and Table 3 to make the main findings more easily accessible directly from the tables.

We believe these revisions strengthen the manuscript and improve its clarity and completeness.

---

### Decision · Action_Editor_2r3U · 2026-02-22

**Recommendation:** Reject

**Additional Comments:**

Should the authors decide to submit a major revision at a later time, they are strongly encouraged to comprehensively address all issues raised by the reviewers, with particular emphasis on strengthening the depth and rigor of the experimental analysis.

**Audience:**

Yes

**Audience Explanation:**

This work is likely to be of interest to a substantial portion of TMLR’s audience, particularly researchers working on image understanding & retrieval, vision and language alignment, large models, and representation learning, etc.

**Claims And Evidence:**

No

**Claims Explanation:**

This work seeks to investigate how perceptual similarity metrics produced by existing models align with human perception.

The reviewers acknowledge several merits of the work, including the introduction of a comprehensive benchmark, extensive experimental evaluation, useful insights, and the practical relevance of the problem.

At the same time, significant concerns were raised. These include the lack of principles in the benchmark design, empirical and superficial analysis, insufficient insight into the underlying causes, conflation between improvements due to data versus method, and a limited discussion of limitations. Notably, two reviewers answered “No” to the question of whether the claims are supported by accurate, convincing, and clear evidence.

In their rebuttal, the authors provided detailed responses and addressed some of the concerns. However, one reviewer indicated that several issues remain unresolved, including terminology inconsistencies, the need for a more substantive limitations section, and the lack of statistical significance analysis. Another reviewer maintained that the responses did not adequately address the raised concerns. The final recommendations were Leaning Reject, Leaning Accept, and Leaning Accept.

After carefully reviewing the manuscript, the reviews, and the author responses, the AE acknowledges the strengths identified by the reviewers, particularly the value of the benchmark. However, the AE concurs with the concerns regarding the depth and rigor of the analysis. In several instances, claims are drawn without sufficient theoretical or experimental analysis. Consequently, the AE cannot conclude that the claims are substantiated by sufficiently rigorous and convincing evidence. Taking all factors into consideration, the AE is unable to recommend acceptance of this submission in its current form.

**Resubmission Of Major Revision:**

The authors may consider submitting a major revision at a later time.